# MOTIONCLONE: TRAINING-FREE MOTION CLONING FOR CONTROLLABLE VIDEO GENERATION

**Pengyang Ling**[1,4*]  **Jiazi Bu**[2,4*]  **Pan Zhang**[4†]  **Xiaoyi Dong**[4]
**Yuhang Zang**[4]  **Tong Wu**[3]  **Huaian Chen**[1]  **Jiaqi Wang**[4]  **Yi Jin**[1†]
[1]University of Science and Technology of China  [2]Shanghai Jiao Tong University
[3]The Chinese University of Hong Kong    [4]Shanghai AI Laboratory
https://github.com/LPengYang/MotionClone

## ABSTRACT

Motion-based controllable video generation offers the potential for creating captivating visual content. Existing methods typically necessitate model training to encode particular motion cues or incorporate fine-tuning to inject certain motion patterns, resulting in limited flexibility and generalization. In this work, we propose **MotionClone**, a training-free framework that enables motion cloning from reference videos to versatile motion-controlled video generation, including text-to-video and image-to-video. Based on the observation that the dominant components in temporal-attention maps drive motion synthesis, while the rest mainly capture noisy or very subtle motions, MotionClone utilizes sparse temporal attention weights as motion representations for motion guidance, facilitating diverse motion transfer across varying scenarios. Meanwhile, MotionClone allows for the direct extraction of motion representation through a single denoising step, bypassing the cumbersome inversion processes and thus promoting both efficiency and flexibility. Extensive experiments demonstrate that MotionClone exhibits proficiency in both global camera motion and local object motion, with notable superiority in terms of motion fidelity, textual alignment, and temporal consistency.

## 1 INTRODUCTION

Video generations that align with human intentions and produce high-quality outputs has recently attracted significant attention, particularly with the rise of mainstream text-to-video (Guo et al., 2023b; Blattmann et al., 2023b; Chen et al., 2024) and image-to-video (Guo et al., 2023a; Blattmann et al., 2023a; Dai et al., 2023) diffusion models. Despite the substantial progress witnessed in conditional image generation, the domain of video generation presents unique challenges, primarily due to the complexities introduced by motion synthesis. Incorporating additional motion control not only mitigates the ambiguity inherent in video synthesis for superior motion modeling but also enhances the manipulability of the synthesized content for customized creations.

In the realm of video generation that is steered by motion cues, pioneering methodologies can be generally classified into two principal strategies: one that leverages the dense depth or sketch of reference videos (Wang et al., 2024; Jeong & Ye, 2023; Guo et al., 2023a), and another that relies on motion trajectories (Wang et al., 2023b; Yin et al., 2023; Niu et al., 2024). The former methodology typically involves the integration of a pre-trained model to extract motion cues at the pixel level. Despite achieving highly aligned motion, these dense motion cues can be intricately entangled with the structural elements of the reference videos, impeding their transferability in novel scenarios. The latter trajectory-based methodology, by contrast, provides a more user-friendly approach for capturing broader object movements but struggles to delineate finer, localized motions such as head turns or hand raises. Additionally, both methodologies typically entail model training to encode particular motion cues, implying suboptimal generation when applied outside the trained domain. Such limitation is also observed in approaches relying on fine-tuning (Jeong et al., 2023; Zhao et al., 2023), which aim to fit the motion patterns of certain videos.

---

*Equal contribution. † Corresponding author.

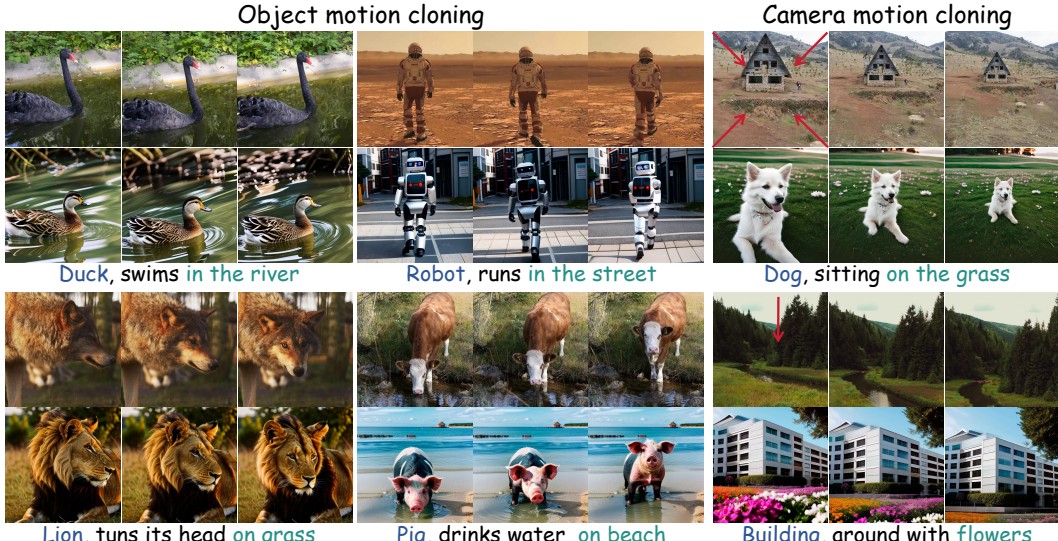

Figure 1: **Motion cloning in varying scenarios.** Given a reference video, **MotionClone** can clone the contained motion into novel scenarios with excellent prompt-following ability, without motion-specific fine-tuning. The red arrows indicate the motion direction.

In this work, we introduce MotionClone, a novel training-free framework designed to clone motions from reference videos for controllable video generation. Diverging from traditional approaches involving tailored training or fine-tuning, MotionClone employs the commonly used temporal-attention mechanism within video generation models to capture the motion in reference videos. This strategy effectively renders detailed motion while concurrently preserving minimal interdependencies with the structural components of the reference video, offering flexible motion cloning in varying scenarios, as shown in Fig. 1. To be specific, it is observed that the dominant components in temporal-attention weights significantly drive motion synthesis, while the rest mainly refer to noisy or very subtle motions. When the whole temporal-attention is applied uniformly across the model, the majority of temporal-attention weights can overshadow the motion guidance, consequently resulting in the suppression of the primary motion. Therefore, we propose to leverage the principal components of the temporal-attention weights as motion representation, which serves as motion guidance that overlooks noisy or less significant motions and concentrates on the primary motion, thus substantially enhancing the fidelity of motion cloning. Moreover, it has been demonstrated that the motion representation extracted from a certain denoising step holds effective guidance across all time steps, offering high efficiency without the burden of cumbersome video inversion. Furthermore, MotionClone is compatible with a range of video generation tasks, including text-to-video (T2V) and image-to-video (I2V), highlighting its versatility and broad applicability.

In summary, (1) we propose **MotionClone**, a novel motion-guided video generation framework that enables training-free motion cloning from given reference videos; (2) we design a primary motion control strategy to perform substantial motion guidance over sparse temporal attention map, allowing for efficient motion transfer across scenarios; (3) we validate the effectiveness and versatility of MotionClone in various video generation tasks, in which extensive experiments demonstrate its proficiency in both global camera motion and local object action, with notable superiority in terms of motion fidelity, text alignment, and temporal consistency.

## 2 RELATED WORK

### 2.1 TEXT-TO-VIDEO DIFFUSION MODELS

Equipped with sophisticated text encoders (Radford et al., 2021; Zhang et al., 2024), a great breakthrough has been achieved in the realm of text-to-image (T2I) generation (Gu et al., 2022; Nichol et al., 2021; Rombach et al., 2022; Podell et al., 2023), which sparks the enthusiasm for advanced text-to-video (T2V) models (Blattmann et al., 2023b; Wang et al., 2023a; Chen et al., 2023a; 2024; Guo et al., 2023b). Notably, VideoLDM (Blattmann et al., 2023b) introduces a motion module that

utilizes 3D convolutions and temporal attention to capture frame-to-frame correlations. In a novel approach, AnimateDiff (Guo et al., 2023b) enhances a pre-trained T2I diffusion model with motion modeling capabilities. This is achieved by fine-tuning a series of specialized temporal attention layers on extensive video datasets, allowing for a harmonious fusion with the original T2I generation process. To address the challenge of data scarcity, VideoCraft2 (Chen et al., 2024) suggests an innovative strategy of learning motion from low-quality videos (Bain et al., 2021) while simultaneously learning appearance from high-quality images (Sun et al., 2024). Despite these advancements, there remains a significant disparity in the quality of generated content between the available T2V models and their sophisticated T2I counterparts, primarily due to the intricate nature of diverse motions and the limited availability of high-quality video data. In this work, a motion guidance strategy is developed, which ingeniously incorporates motion cues from given videos to ease the challenges of motion modeling, yielding more realistic and coherent video sequences, without model fine-tuning.

## 2.2 CONTROLLABLE VIDEO GENERATION

Building on the success of controllable image generation through the integration of additional conditions (Zhang et al., 2023; Kim et al., 2023; Li et al., 2023; Qin et al., 2023; Huang et al., 2023), a multitude of studies (Chen et al., 2023a; Yin et al., 2023; Dai et al., 2023; Ma et al., 2024; Blattmann et al., 2023a) have endeavored to introduce diverse control signals for versatile video generation. These include control over the first video frame (Chen et al., 2023a), motion trajectory (Yin et al., 2023), motion region (Dai et al., 2023), and motion object (Ma et al., 2024). Furthermore, in pursuit of high-quality video customization, several studies delve into reference-based video generation, leveraging the motion from an existing real video to direct the creation of new video content. A straightforward solution developed in Wang et al. (2024); Esser et al. (2023); Xing et al. (2024), involves the direct integration of frame-wise depth maps or canny maps to regularize motion. However, this approach inadvertently introduces motion-independent features, such as structures in static areas, which can disrupt the alignment of the resulting video appearance with new text. To address this issue, motion-specific fine-tuning frameworks, as explored in (Zhao et al., 2023; Jeong et al., 2023), have been developed to extract a distinct motion pattern from a single video or a collection of videos with identical motion. While holding promise, these methods are subject to complex training processes and potential model degradation. To address this, we present a novel motion cloning scheme, which extracts temporal correlations from existing videos as explicit motion clues to guide the generation of new video content, providing a plug-and-play motion customization solution.

## 2.3 ATTENTION FEATURE CONTROL

Attention mechanisms have been confirmed as vital for high-quality content generation. Prompt2Prompt (Hertz et al., 2022) illustrates that cross-attention maps are instrumental in dictating the spatial layout of synthesized images. This observation subsequently motivates serious work in semantic preservation (Chefer et al., 2023), multi-object generation (Ma et al., 2023; Xiao et al., 2023), and video editing (Liu et al., 2023). AnyV2V (Ku et al., 2024) reveals dense injection of both CNN and attention features facilitates improved alignment with source videos in video editing. FreeControl (Mo et al., 2023) highlights that the feature space within self-attention layers encodes structural image information, facilitating reference-based image generation. While previous methods mainly concentrate on spatial attention layers, our work uncovers the untapped potential of temporal attention layers for effective motion guidance, enabling flexible motion cloning.

## 3 MOTIONCLONE

In this section, we first introduce video diffusion models and temporal attention mechanisms. Then we explore the potential of primary control over sparse temporal attention maps for substantial motion guidance. Subsequently, we elaborate on the proposed MotionClone framework, which performs motion cloning by deliberately manipulating temporal attention weights.

### 3.1 PRELIMINARIES

**Diffusion sampling.** Following pioneering work (Rombach et al., 2022), video diffusion models encode a input video $x$ into latent representation $z_0 = \mathcal{E}(x)$ by using a pre-trained encoder $\mathcal{E}(\cdot)$. To

Reference video · Prompt: A cat plays in the forest · Generated videos from same initial noise · Reference video · Prompt: A tank runs in the desert · Generated videos from same initial noise

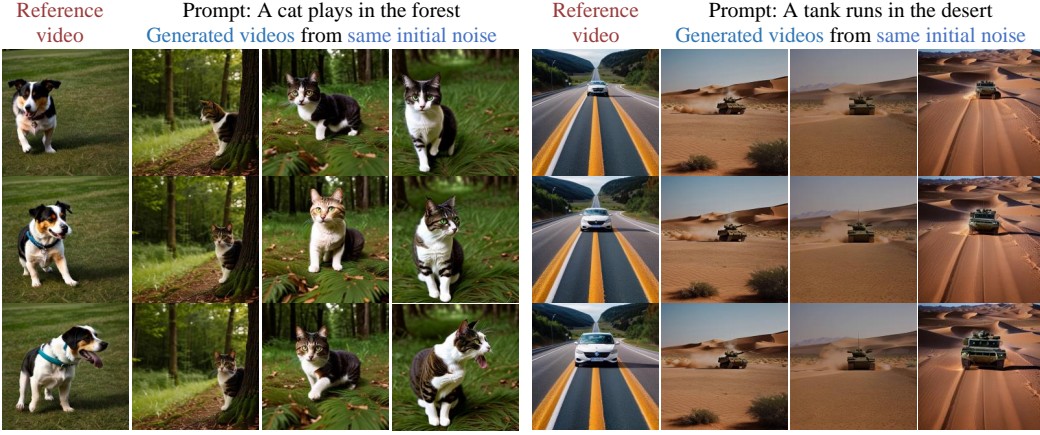

w/o control · Plain control · Primary control · w/o control · Plain control · Primary control

Figure 2: **Comparision of plain control and primary control over temporal attention map.** Leveraging temporal attention maps derived from reference videos to guide video generation. *Plain control* refers to a rudimentary approach whereby all weights are uniformly applied. *Primary control* only applies constraint to the sparse temporal attention map.

enable video distribution learning, diffusion model $\epsilon_\theta$ is encouraged to estimate noise component $\epsilon$ from noised latent $z_t$ that follows time-dependent scheduler (Ho et al., 2020), i.e.,

$$\mathcal{L}(\theta) = \mathbb{E}_{\mathcal{E}(x),\epsilon \in \mathcal{N}(0,1),t \sim \mathcal{U}(1,T)} \left[ \| \epsilon - \epsilon_\theta(z_t, c, t) \|_2^2 \right], \quad (1)$$

where $t$ is the time step, and $c$ is the condition signal such as text or image. In the inference phase, the sampling process commences with a standard Gaussian noise. The sampling trajectory, however, can be adjusted by incorporating guidance for extra controllability. This is typically achieved by customized energy function $g(z_t, y, t)$ with label $y$ indicating guidance direction, i.e.,

$$\hat{\epsilon_\theta} = \epsilon_\theta(z_t, c, t) + s(\epsilon_\theta(z_t, c, t) - \epsilon_\theta(z_t, \phi, t)) + \lambda \sqrt{1 - \bar{\alpha}_t} \nabla_{z_t} g(z_t, y, t), \quad (2)$$

where $\epsilon_\theta(z_t, \phi, t)$ is the classifier-free guidance (Ho & Salimans, 2022), $\phi$ denotes the unconditional class identifier (such as null text for textual condition), $s$ and $\lambda$ are guidance weights, and the term $\sqrt{1 - \bar{\alpha}_t}$ is used to convert the gradient of energy function $g(\cdot)$ into noise prediction, in which $\sqrt{\bar{\alpha}_t}$ is the hyperparameter of noise schedule, i.e., $z_t = \sqrt{\bar{\alpha}_t} z_0 + \sqrt{1 - \bar{\alpha}_t} \epsilon$. During sampling, the gradient generated by energy function $g(\cdot)$ indicates the direction toward generation target.

**Temporal attention.** In video motion synthesis, temporal attention mechanism is broadly applied to establish correlation across frames. Given a $f$-frame video feature $f_{in} \in \mathbb{R}^{b \times f \times c \times h \times w}$ where $b$ denotes batch size, $c$ denotes channel number, $h$ and $w$ are spatial resolution, temporal attention first reshapes it into 3D tensor $f'_{in} \in \mathbb{R}^{(b \times h \times w) \times f \times c}$ by merging the spatial dimensions into the batch size. Subsequently, it executes self-attention along the frame axis, which can be expressed as:

$$f_{out} = Attention(Q(f'_{in}), K(f'_{in}), V(f'_{in})), \quad (3)$$

where $Q(\cdot)$, $K(\cdot)$, and $V(\cdot)$ are projection layers. Correspondingly, the attention map is labeled as $\mathcal{A} \in \mathbb{R}^{(b \times h \times w) \times f \times f}$, which captures the temporal relation for each pixel feature.

## 3.2 OBSERVATION

Since temporal attention mechanism governs the motion in the generated video, videos with similar temporal attention maps are expected to share similar motion characteristics. To investigate this hypothesis, we manipulate the sampling trajectory by aligning the temporal attention maps of the generated video with those from a reference video. As depicted in Fig. 2, simply enforcing alignment on the entire temporal attention map (plain control) can only partly restore coarse motion patterns of reference videos, such as the gait of a cat and the directional movement of a tank, demonstrating limited motion alignment. We postulate that this is because not all temporal attention weights are essential for motion synthesis, with some reflecting scene-specific noise or extremely small motions. Indiscriminate alignment with the entire temporal attention maps dilutes critical motion guidance, resulting in suboptimal motion cloning in novel scenarios. As evidence, primary control over the sparse temporal attention map significantly boosts motion alignment, which can be attributed to the emphasis on motion-related cues and the disregard of motion-irrelevant factors.

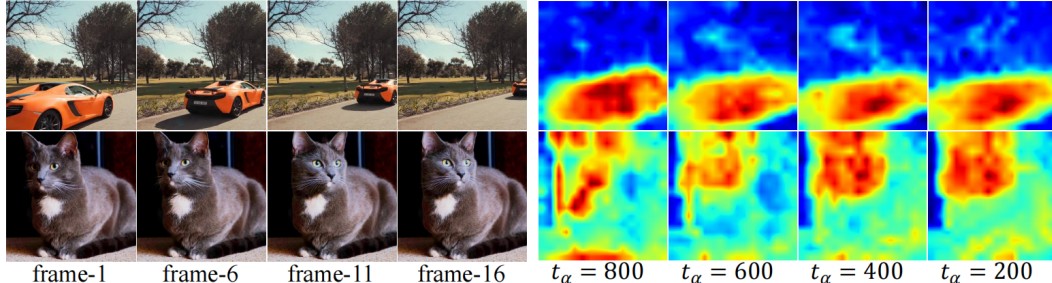

| frame-1 | frame-6 | frame-11 | frame-16 | $t_\alpha = 800$ | $t_\alpha = 600$ | $t_\alpha = 400$ | $t_\alpha = 200$ |

Figure 3: **Visualization of motion representation.** The mean intensity of $\mathcal{L}^{t_\alpha}$ in frame axis from "up_blocks.1" (resized to the represented resolution) indicates the area and magnitude of motion. This performance encounters decline in complex "head turning" scenario when $t_\alpha = 800$.

### 3.3 MOTION REPRESENTATION

Given a reference video, the corresponding temporal attention map in $t$ denoising step is denoted as $\mathcal{A}^t_{ref} \in \mathbb{R}^{(1 \times h \times w) \times f \times f}$, which satisfies $\sum_{j=1}^{f} [\mathcal{A}^t_{ref}]_{p,i,j} = 1$. The value of $[\mathcal{A}^t_{ref}]_{p,i,j}$ reflects the relation between $i$ frame and $j$ frame in position $p$, and a larger value of $[\mathcal{A}^t_{ref}]_{p,i,j}$ implies a stronger correlation. The motion guidance over temporal attention maps, depicted by energy function $g(\cdot)$, is modeled as:

$$g = \left\| \mathcal{M}^t \cdot (\mathcal{A}^t_{ref} - \mathcal{A}^t_{gen}) \right\|_2^2, \tag{4}$$

where $\mathcal{M}^t$ is the temporal mask for sparse constraint, and $\mathcal{A}^t_{gen}$ is the temporal attention weights of generated videos in time step $t$. Essentially, Eq. 4 promotes motion cloning by forcing $\mathcal{A}^t_{gen}$ close to $\mathcal{A}^t_{ref}$, while $\mathcal{M}^t$ determines the sparsity of constraint, time-dependence $\left\{ \mathcal{A}^t_{ref}, \mathcal{M}^t \right\}$ constitute the motion guidance. Particularly, $\mathcal{M}^t \equiv 1$ refers to the "plain control" that exhibits limited motion transfer capability as illustrated in Fig. 2. Since the value of $\mathcal{A}^t_{ref}$ is indicative of the strength of inter-frame correlation, we propose to obtain the sparse temporal mask according to the rank of $\mathcal{A}^t_{ref}$ value in the temporal axis, i.e.,

$$\mathcal{M}^t_{p,i,j} := \begin{cases} 1, & if \ [\mathcal{A}^t_{ref}]_{p,i,j} \in \Omega^t_{p,i} \\ 0, & otherwise, \end{cases} \tag{5}$$

where $\Omega^t_{p,i} = \{\tau_1, \tau_2, ..., \tau_k\}$ is the subset of index that comprising the top $k$ values in attention map $\mathcal{A}^t_{ref}$ along the temporal axis $j$, and $k$ is a hyper-parameter. Particularly, in the case where $k = 1$, motion guidance focuses solely on the highest activation for each spatial location. Supervised by Eq. 5, motion guidance in Eq. 4 encourages the sparse alignment with the primary component in $\mathcal{A}^t_{ref}$ while ensures spatially even constraint, facilitating a stable and reliable motion transfer.

Despite enabling effective motion cloning, the above scheme has obvious flaws: i) for real reference videos, laborious and time-consuming inversion operation is required for preparing $\mathcal{A}^t_{ref}$; and ii) the considerable size of the time-dependent $\left\{ \mathcal{A}^t_{ref}, \mathcal{M}^t \right\}$ poses significant challenges for large-scale preparation and efficient deployment. Fortunately, it is noted that the representation from certain denoising step can provide substantial and consistent motion guidance in generation process. Mathematically, motion guidance in Eq. 4 can be converted into

$$g = \left\| \mathcal{M}^{t_\alpha} \cdot (\mathcal{A}^{t_\alpha}_{ref} - \mathcal{A}^t_{gen}) \right\|_2^2 = \left\| \mathcal{L}^{t_\alpha} - \mathcal{M}^{t_\alpha} \cdot \mathcal{A}^t_{gen} \right\|_2^2, \tag{6}$$

where $t_\alpha$ denotes certain time step, and $\mathcal{L}^{t_\alpha} = \mathcal{M}^{t_\alpha} \cdot \mathcal{A}^{t_\alpha}_{ref}$. For given reference videos, the corresponding motion representation is denoted as $\mathcal{H}^{t_\alpha} = \{\mathcal{L}^{t_\alpha}, \mathcal{M}^{t_\alpha}\}$, comprising two elements that are both highly temporally sparse. For real reference videos, their $\mathcal{H}^{t_\alpha}$ can be easily derived by directly adding noise to shift them into the noised latent of $t_\alpha$ time step, followed by a single denoising step. This straightforward strategy, impressively, proves to be remarkably effective. As shown in Fig. 3, over a larger range of time steps ($t_\alpha$ from 200 to 600), the mean intensity of $\mathcal{H}^{t_\alpha}$ effectively highlights the region and magnitude of motion. However, it is also observed that $\mathcal{H}^{t_\alpha}$ in early denoising stage ($t_\alpha = 800$) shows some discrepancies with the "head-turning" motion. This can be attributed

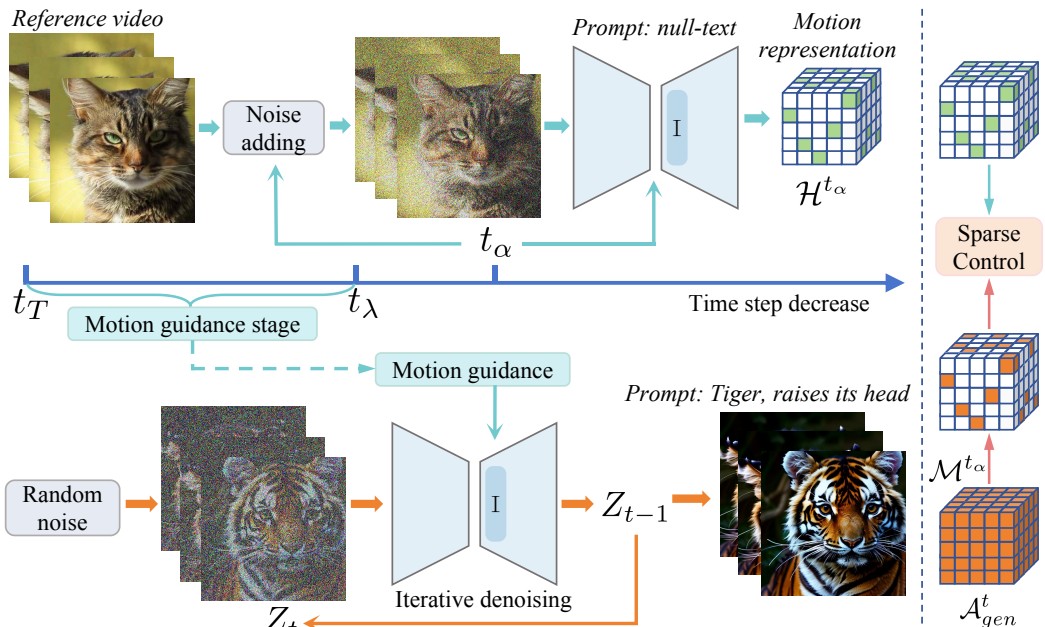

Figure 4: **The pipeline of MotionClone,** in which the motion representation $\mathcal{H}^{t_\alpha}$ extracted from reference videos serves as motion guidance in novel video synthesis.

to the fact that motion synthesis has not yet been fully determined at this early stage. Therefore, we suggest employing the motion-aligned $\mathcal{H}^{t_\alpha}$ from the latter denoising stage to guide motion synthesis in the entire sampling process, facilitating substantial and consistent motion guidance for superior motion alignment.

### 3.4 MOTION GUIDANCE

The pipeline of MotionClone is depicted in Fig. 4. Given a real reference video, the corresponding motion representation $\mathcal{H}^{t_\alpha}$ is obtained by performing a single noise-adding and denoising step. During the video generation process, an initial latent is initialized from a standard Gaussian distribution and subsequently undergoes an iterative denoising procedure via a pre-trained video diffusion model, advised by both classifier-free guidance and the proposed motion guidance. Given that image structure is determined in the early steps of the denoising process (Hertz et al., 2022), whereas motion fidelity primarily depends on the structure of each frame, motion guidance only involves the early denoising steps, allowing for sufficient flexibility for semantic adjustment and thus empowering premium video generation with compelling motion fidelity and precise textual alignment.

## 4 EXPERIMENTS

### 4.1 IMPLEMENTATION DETAILS

In this work, we employ AnimateDiff(Guo et al., 2023b) as the base text-to-video generation model and leverage SparseCtrl (Guo et al., 2023a) for image-to-video and sketch-to-video generator. For given real videos, we apply single denoising in $t_\alpha = 400$ for motion representation extraction. $k = 1$ is adopted for mask in Eq. 5 to facilitate sparse constraint. "null-text" is uniformly used as textual prompt for preparing motion representations, promoting a more convenient video customization. The motion guidance is conducted on temporal attention layers in "up_block.1". The detailed ablations of above setting are represented in 4.6. Guidance weight $s$ and $\lambda$ in Eq. 2 are empirically set as 7.5, and 2000, respectively. For camera motion cloning, the denoising step is configured to 100, in which the motion guidance steps set as 50. For object motion cloning, the denoising step is raised to 300, while applying motion guidance in the early 180 steps.

*Prompt: Teddy bear, on the grass.*

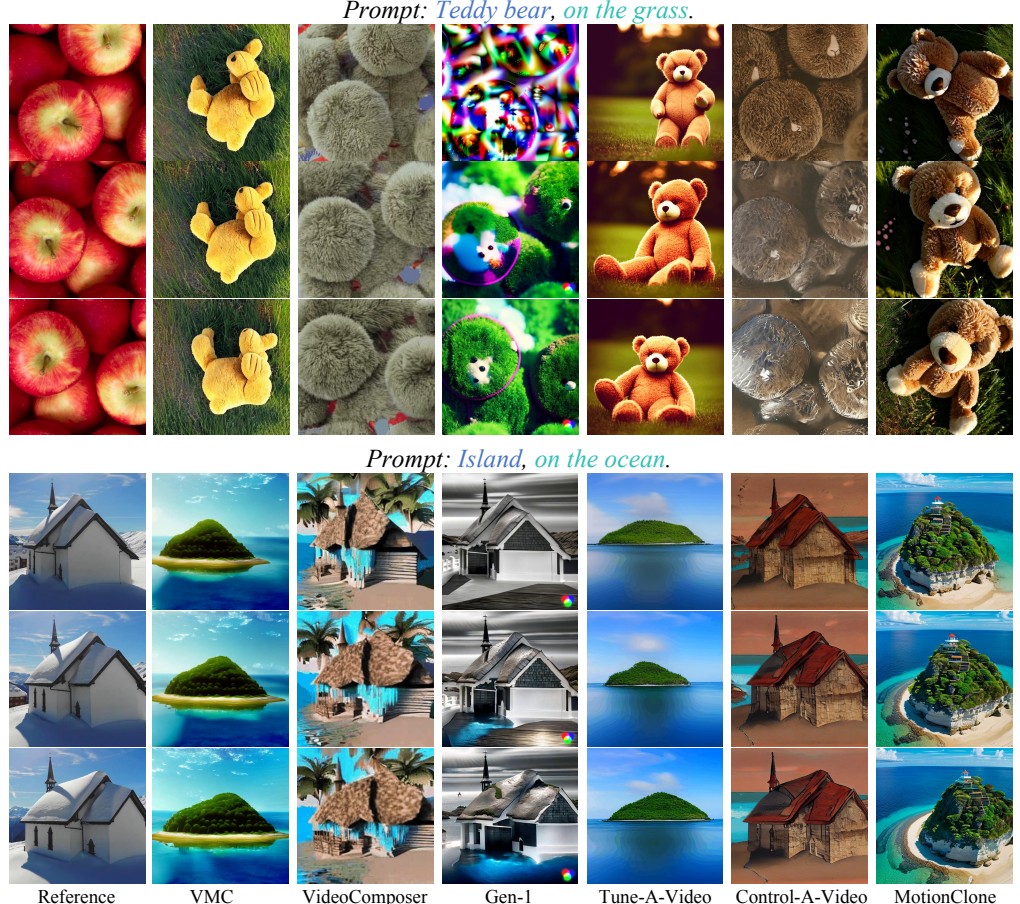

*Prompt: Island, on the ocean.*

| Reference | VMC | VideoComposer | Gen-1 | Tune-A-Video | Control-A-Video | MotionClone |

Figure 5: **Visual comparison in camera motion cloning,** in which **MotionClone** achieves superior textual alignment by better suppressing the original structure.

## 4.2 EXPERIMENTAL SETUP

**Dataset.** For experimental evaluation, 40 real videos sourced from DAVIS (Pont-Tuset et al., 2017) and website are utilized for a thorough analysis, comprising 15 videos with camera motion and 25 videos for object motion. These videos encompass a rich tapestry of motion types and scenarios, ranging from the dynamic motions of animals and humans to the global camera motion.

**Evaluation metrics** For objective evaluation, two commonly used metrics are adopted: i) Textual alignment, which quantifies the congruence with the provided textual prompt. Following previous work (Wang et al., 2024), it is measured by the average CLIP (Radford et al., 2021) cosine similarity between all video frames and text (Jeong et al., 2023); ii) Temporal consistency, the indicator of video smoothness, is quantified by calculating the average CLIP similarity among consecutive video frames. Beyond the scope of objective metrics, a user study is employed for a more nuanced assessment of human preferences in video quality, incorporating two additional criteria: i) motion preservation which evaluates the motion's adherence to the reference video, and ii) appearance diversity which assesses the visual range and diversity in contrast to the reference video. The user study scores are derived from the average ratings provided by 20 volunteers, ranging from 1 to 5.

**Baselines.** For a thorough comparative analysis, various alternative methods have been examined in the comparison, including VideoComposer (Wang et al., 2024), Tune-A-video (Wu et al., 2023), Control-A-Video (Chen et al., 2023b), VMC (Jeong et al., 2023), and Gen-1 (Esser et al., 2023). A detailed description of each method is depicted in the Appendix.

## 4.3 QUALITATIVE COMPARISON

**Camera motion cloning.** As shown in Fig. 5, the "clockwise rotation" and "view switching" motion present a significant challenge. VMC and Tune-A-Video generate scenes with acceptable textual

*Cat, moves its head, in the bedroom.*

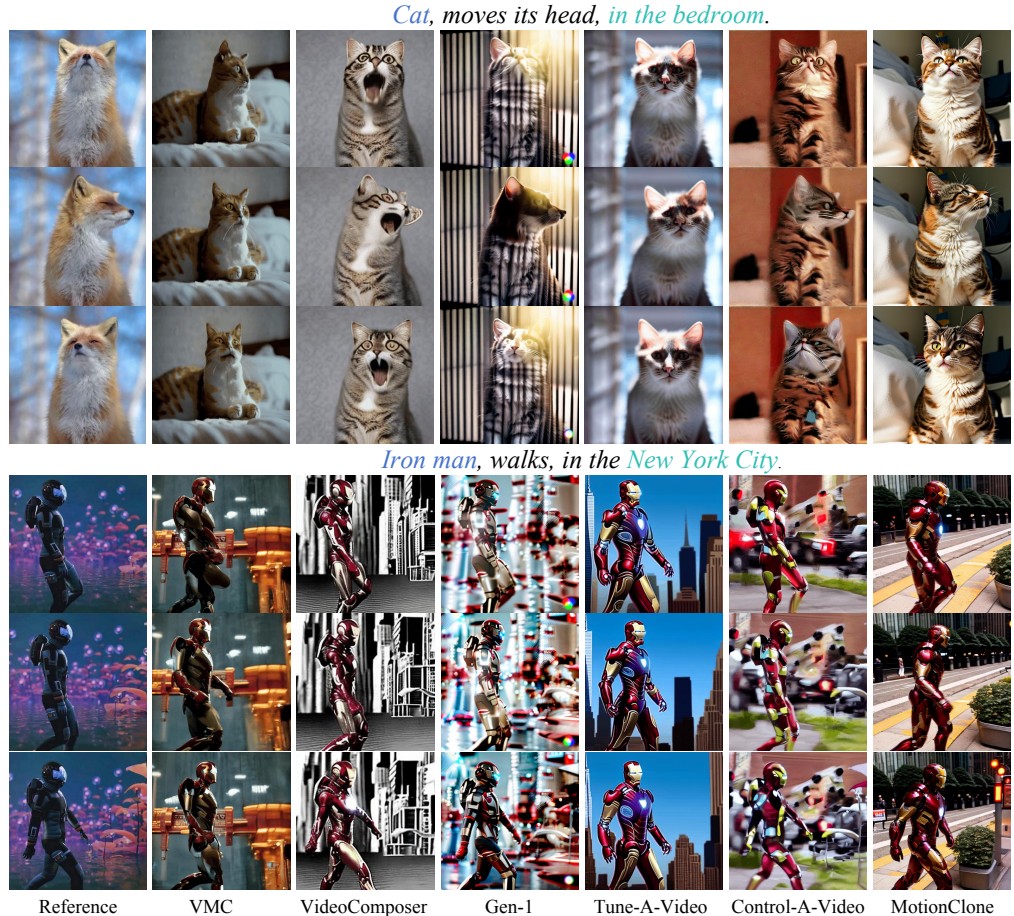

*Iron man, walks, in the New York City.*

| Reference | VMC | VideoComposer | Gen-1 | Tune-A-Video | Control-A-Video | MotionClone |

Figure 6: **Visual comparison in object motion cloning,** in which **MotionClone** performs preferable motion fidelity with improved prompt-following ability.

Table 1: **Quantitative comparison** by using automotive metrics and user study.

| Method | VMC | VideoComposer | Gen-1 | Tune-A-Video | Control-A-Video | **MotionClone** |
|---|---|---|---|---|---|---|
| Textual Alignment | 0.3134 | 0.2854 | 0.2462 | 0.3002 | 0.2859 | **0.3187** |
| Temporal Consistency | 0.9614 | 0.9577 | 0.9563 | 0.9351 | 0.9513 | **0.9621** |
| Motion Preservation | 2.59 | 3.28 | 3.50 | 2.44 | 3.33 | **3.69** |
| Appearance Diversity | 3.51 | 3.23 | 3.25 | 3.09 | 3.27 | **4.31** |
| Textual Alignment | 3.79 | 2.71 | 2.80 | 3.04 | 2.82 | **4.15** |
| Temporal Consistency | 2.85 | 2.79 | 3.34 | 2.28 | 2.81 | **4.28** |

alignment but exhibit deficiencies in motion transfer. The outputs from VideoComposer, Gen-1, and Control-A-Video are notably unrealistic, which can be attributed to the dense integration of the structural elements from the original videos. Conversely, MotionClone demonstrates superior text alignment and motion consistency, thereby suggesting its superior video motion transfer capabilities within global camera motion scenarios.

**Object motion cloning.** Beyond the scope of camera motion, the proficiency in handling local object motions has been rigorously validated. As evidenced by Fig. 6, VMC falls short in matching motion with the source videos. Videocomposer appears to generate grayish colors with limited prompt-following ability. Gen-1 is inhibited by the original videos' structure. Tune-A-Video struggles with capturing detailed body motions, while Control-A-Video cannot maintain a faithful appearance. In contrast, MotionClone stands out in scenarios with localized object motions, enhancing motion accuracy and improved text alignment.

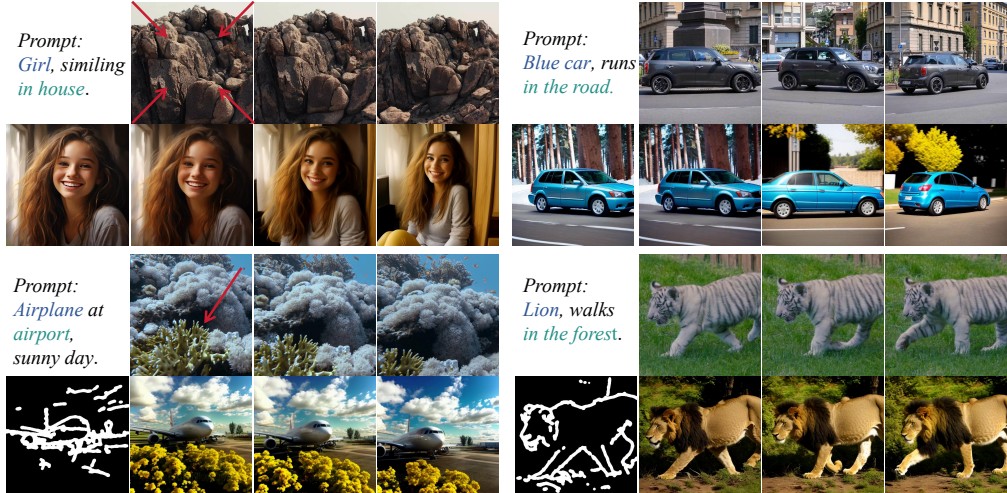

Figure 7: **MotionClone also supports I2V and sketch-to-video,** facilitating versatile applications. The red arrows indicate the motion direction.

## 4.4 QUANTITATIVE COMPARISON

The quantitative comparison on 40 real videos with various motion patterns is outlined in Tab. 1. It is observed that MotionClone gains competitive scores in both textual alignment and temporal consistency. Moreover, MotionClone has outperformed its rivals in motion preservation, appearance diversity, temporal consistency, and textual alignment in human preference tests, underscoring its ability to produce visually compelling outcomes.

## 4.5 VERSATILE APPLICATION

Beyond T2V, MotionClone is also compatible with I2V and sketch-to-video. As shown in Fig. 7, by incorporating the first frame or a sketch image as an additional condition, MotionClone achieves impressive motion transfer while aligning with the specified condition, underscoring its significant potential for a wide range of applications.

## 4.6 ABLATION AND ANALYSIS

**Choice of $k$.** $k$ determines the mask in Eq. 5 and thus impacts the sparsity of motion constraint. As illustrated in Fig. 8, a lower $k$ value helps better primary motion alignment, attributed by the enhanced elimination of scene-specific noise and subtle motion.

**Choice of $t_\alpha$.** The value of $t_\alpha$ determines diffusion feature distribution used for preparing motion representations. As shown in Fig. 8, an excessively large $t_\alpha = 800$ causes substantial loss of motion information due to excessive noise injection, while $t_\alpha \in \{200, 400, 600\}$ can all achieve a certain degree of motion alignment, implying the robustness of $t_\alpha$. In this work, we chose $t_\alpha = 400$ as the default value as it typically yields appealing motion cloning in our experiments.

**Choice of temporal attention block.** Fig. 9 illustrates the results with motion guidance applied in different blocks. It is observed that "up_block.1" stands out for its superior motion manipulation capabilities while safeguarding visual quality, underscoring its dominant role in motion synthesis.

**Does precise prompt help ?** During motion representation preparation procedure, few differences arise when using tailored prompts regarding video content, as represented in Fig. 9. We speculate that motion-related information is effectively preserved in diffusion features at $t_\alpha = 400$, thereby diminishing the significance of the precise prompt.

**Does video inversion help ?** Video inversion can provide time-dependence $\left\{\mathcal{A}_{ref}^t, \mathcal{M}^t\right\}$ for Eq. 4 and certain time step $\{\mathcal{L}^{t_\alpha}, \mathcal{M}^{t_\alpha}\}$ for Eq. 6, but entails considerable time costs. As shown in Fig. 9 (Inversion_1 *vs.* Inversion_2), $\{\mathcal{L}^{t_\alpha}, \mathcal{M}^{t_\alpha}\}$ outperforms $\left\{\mathcal{A}_{ref}^t, \mathcal{M}^t\right\}$ due to the consistent motion

*Promt: Woman, walks in the mall.*     *Promt: Shark, swims on the ocean.*

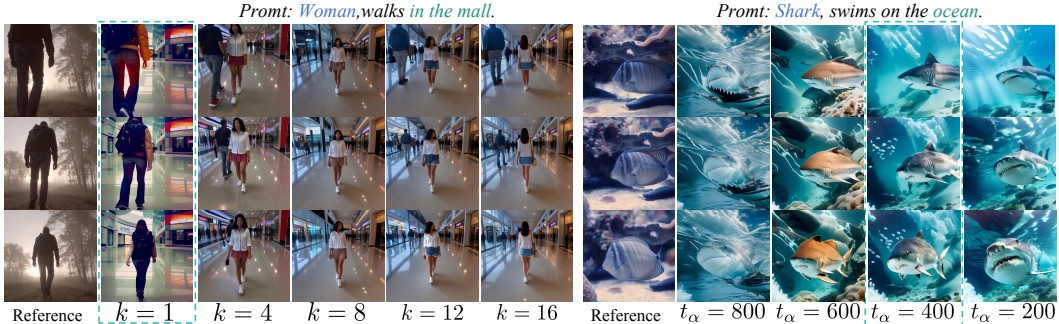

Reference  $k = 1$  $k = 4$  $k = 8$  $k = 12$  $k = 16$  Reference  $t_\alpha = 800$  $t_\alpha = 600$  $t_\alpha = 400$  $t_\alpha = 200$

Figure 8: **Influence of different $k$ value and different time step $t_\alpha$.**

*Prompt: Dog, walks in the street.*     *Prompt: Spider Man, turns his head.*

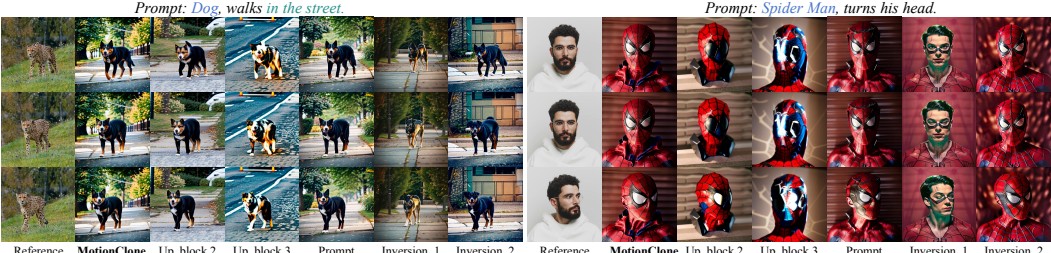

Reference  **MotionClone**  Up_block.2  Up_block.3  Prompt  Inversion_1  Inversion_2   Reference  **MotionClone**  Up_block.2  Up_block.3  Prompt  Inversion_1  Inversion_2

Figure 9: **Influence of different attention blocks, precise prompt, and DDIM inversion.** "Prompt" denotes motion representation involves precise prompt ("Leopard, walks in the forest" for the left case and "Man, turns his head." for the right case); "Inversion_1" represents time-dependence $\left\{ \mathcal{A}_{ref}^t, \mathcal{M}^t \right\}$ from DDIM inversion; "Inversion_2" indicates $\{ \mathcal{L}^{t_\alpha}, \mathcal{M}^{t_\alpha} \}$ from DDIM inversion.

guidance from the same representation. Meanwhile, there is not obvious quality difference regarding whether DDIM inversion is applied (MotionClone *vs.* Inversion_2). We leave how to perform better diffusion inversion for enhanced motion cloning for further work.

## 4.7 LIMITATION

Given that MotioClone is conducted in latent space, the spatial resolution of diffusion features in temporary attention is significantly lower than that of input videos, thus MotionClone struggles in local subtle motion, such as winking, as shown in Fig. 10. Additionally, when multiple moving objects overlap, MotionClone risks quality dropping, attributing that coupled motion raises the difficulty of motion cloning.

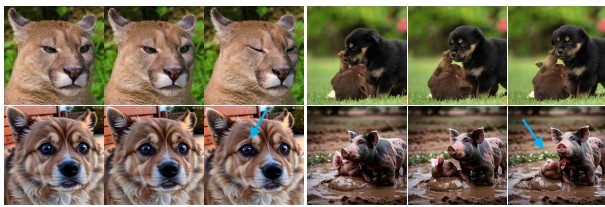

*Prompt: A dog is winking.*     *Prompt: Pigs, play in the mud.*

Figure 10: MotionClone struggles to handle local subtle motion and overlapping motion.

## 5 CONCLUSION

In this work, we observe that the temporal attention layers embedded within video generation models harbor substantial representational capacities pertinent to video motion transfer. Motivated by these findings, we introduce a training-free method dubbed MotionClone for motion cloning. Leveraging sparse temporal attention weights as motion representations, MotionClone facilitates motion guidance by promoting primary motion alignment, enabling diverse motion transfer across different scenarios. Employing a real reference video, MotionClone demonstrates its capability to preserve motion fidelity robustly while concurrently assimilating novel textual semantics. Furthermore, MotionClone demonstrates efficiency by avoiding cumbersome inversion processes and offers versatility across various video generation tasks, establishing itself as a highly adaptable and efficient tool for motion customization.

ACKNOWLEDGMENTS

This project is funded by Anhui Provincial Natural Science Foundation 2408085J024; National Natural Science Foundation of China 62401532; and Anhui Provincial Key Research and Development Plan 202304a05020072. This project is funded in part by Shanghai Artificial Intelligence Laboratory, Shanghai Innovation Institute, the National Key RD Program of China (2022ZD0160201).

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

# A APPENDIX

## A.1 BASELINE DESCRIPTION

Among the compared methods, **VideoComposer** (Wang et al., 2024) creates videos by extracting specific features such as frame-wise depth or canny maps from existing videos, achieving a compositional approach to controllable video generation. **Gen-1** (Esser et al., 2023) leverages the original structure of reference videos to generate new video content, akin to video-to-video translation. **Tune-A-Video** expands the spatial self-attention of pre-trained text-to-image models into spatio-temporal attention, and then fine-tuning it for motion-specific generation. **Control-A-Video** (Chen et al., 2023b) incorporates the first video frame as an additional motion cue for customized video generation. **VMC** (Jeong et al., 2023) aims to distill motion patterns by fine-tuning the temporal attention layers in a pre-trained text-to-video diffusion model.

## A.2 ABLATION STUDY OF TEMPORAL ATTENTION IN DOWN BLOCK

We have further evaluated the performance of applying motion guidance in down blocks, the qualitative results are represented in Fig. 11. It can be observed that "up_block.1" showcases superiority in motion customization, which performs better motion alignment while enabling effective semantic preservation. We conjecture this is because the video motion is determined by the structure of each frame, which is mainly modeled in the "up_block.1", as validated by previous image structure customization work Mo et al. (2023).

## A.3 COMPARISON WITH MORE METHODS

We have also compared the proposed method with more methods, including MOFT Xiao et al. (2024), Space-Time Diffusion Features Yatim et al. (2024), UniEdit Bai et al. (2024), and MotionMaster Hu et al. (2024). The generated videos of different methods regarding camera motion cloning and object motion cloning are presented in Fig. 12 and Fig. 13, respectively. It is observed that the proposed MotionClone achieves better motion alignment while performing effective appearance preservation, thus offering superior motion customization. Moreover, it is worth mentioning that all of the compared methods require time-consuming DDIM inversion, which implies high computation costs in practical application. In comparison, the proposed MotionClone can extract motion representation from a single denoising step with a unified null-text prompt, offering both efficiency and flexibility.

## A.4 APPLICATION IN DiT-BASED ARCHITECTURE

We have validated the potential of the proposed MotionClone in the latest DiT-based T2V model, i.e., CogVideoX-2B Yang et al. (2024), in which MotionClone demonstrates inspiring effectiveness in training-free motion customization by only using motion representation from a single denoising step, offering both generalizability and flexibility. The results are represented in Fig. 14, and the detailed implementations are listed below. The video synthesis stage in Diffusion DiT involves the collaboration between textual token $s_{text} \in \mathbb{R}^{(1 \times n_1 \times c)}$ and image token $s_{img} \in \mathbb{R}^{(1 \times n_2 \times c)}$, in which $n_1$ and $n_2$ are token number, and $c$ is channel number. During video synthesis, CogVideoX leverages a full attention module for complete relation modeling, in which the resolution of the vanilla attention map thus reaches $1 \times (n_1 + n_2) \times (n_1 + n_2)$. Since we focus on the dependence between image tokens for motion transfer across scenarios, we introduce an auxiliary attention map to better describe inter-frame dependencies. Specifically, within the self-attention module, the image token is reshaped into $s_{img} \in \mathbb{R}^{(1 \times n_h \times h \times w \times f \times c)}$, in which $n_h$ is the head number of multi-head attention, $h$ and $w$ are spatial resolution, and $f$ is frame number, the corresponding attention map $A_{aux} \in \mathbb{R}^{1 \times n_h \times h \times w \times f \times f}$ can be obtained by using self-attention in the last frame axis. Following the operation in Diffusion U-Net, MotionClone applies sparse constraints $A_{aux}$ according to the rank of their values. For given real reference videos, we use null-text as the default prompt for preparing motion representation to facilitate user-friendly deployment.

### A.5 MORE GENERATED RESULTS

A broader array of generated content is displayed to validate the versatile generation capability. As shown in Figs. 15- 18, **MotionClone** is able to adeptly extract motion cues from a diverse range of existing videos and thus enables the creation of content that is both prompt-aligned and motion-preserved, showcasing its robust motion cloning capabilities. For a better demonstration of MotionClone, we highly recommend viewing the video file in the supplementary material.

### A.6 BROADER IMPACT

The development of MotionClone, a novel training-free framework for motion-based controllable video generation, carry distinct societal implications, both beneficial and challenging.

On the positive side, MotionClone's capability to efficiently clone motions from reference videos while ensuring high fidelity and textual alignment opens new avenues in numerous fields. In the realm of digital content creation, film and media professionals can utilize this technology to streamline the production process, enhance narrative expressions, and create more engaging visual experiences without extensive resource commitments. Furthermore, in the educational sector, instructors and content creators can leverage this innovation to produce customized instructional videos that incorporate precise motions aligned with textual descriptions, potentially increasing engagement and comprehension among students. This could be particularly transformative for subjects where demonstration of physical actions or processes plays a crucial role, such as in sports training or scientific experiments.

On the negative side, the power of MotionClone to generate realistic videos based on text and existing motion cues raises concerns about its potential misuse, including the creation of deepfakes or misleading media content. Such applications can undermine trust in media, affect public opinion through the dissemination of false information, and infringe on personal rights and privacy. Moreover, the ease of generating convincing videos might enable the proliferation of propaganda or harmful content that can have widespread negative implications on society.

In conclusion, while MotionClone presents significant advancements in the field of AI-driven video generation, it is imperative that these technologies are developed and utilized with a conscious commitment to ethical standards and regulatory oversight. Promoting transparency in AI-generated content, establishing clear usage guidelines, and fostering an open dialogue about the capabilities and ethics of such technologies are crucial steps in ensuring that the benefits of MotionClone are realized while its risks are effectively mitigated. This involves collaborative efforts among technologists, policymakers, industry stakeholders, and the broader public to steer the responsible development and application of AI-driven media tools.

Reference  Down_block.1  Down_block.2  Down_block.3  **Ours**

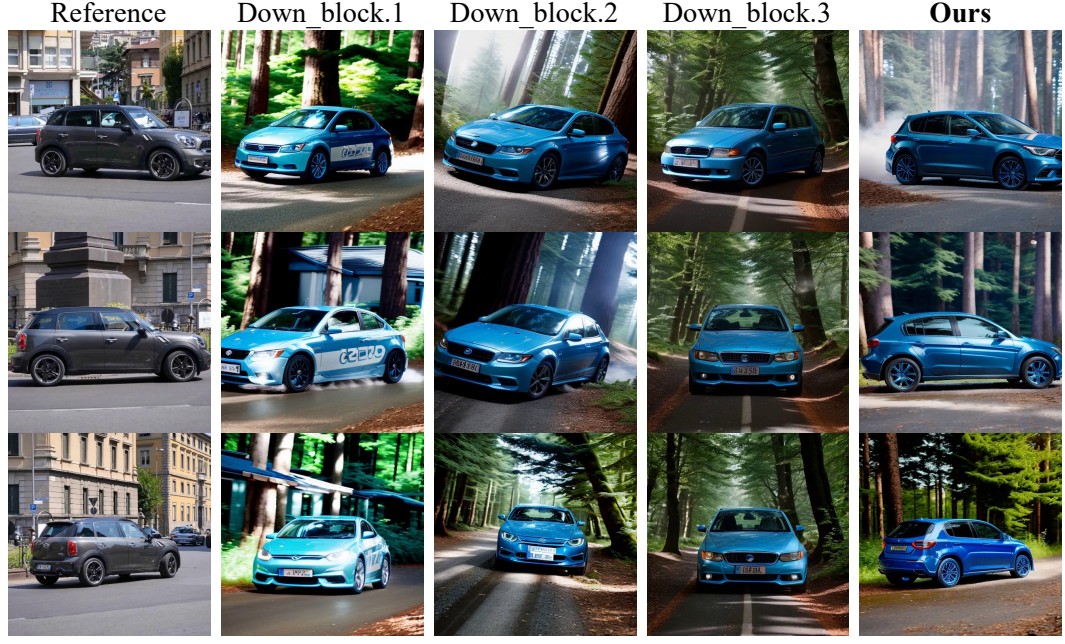

*A blue car, running, in the forest, ...*

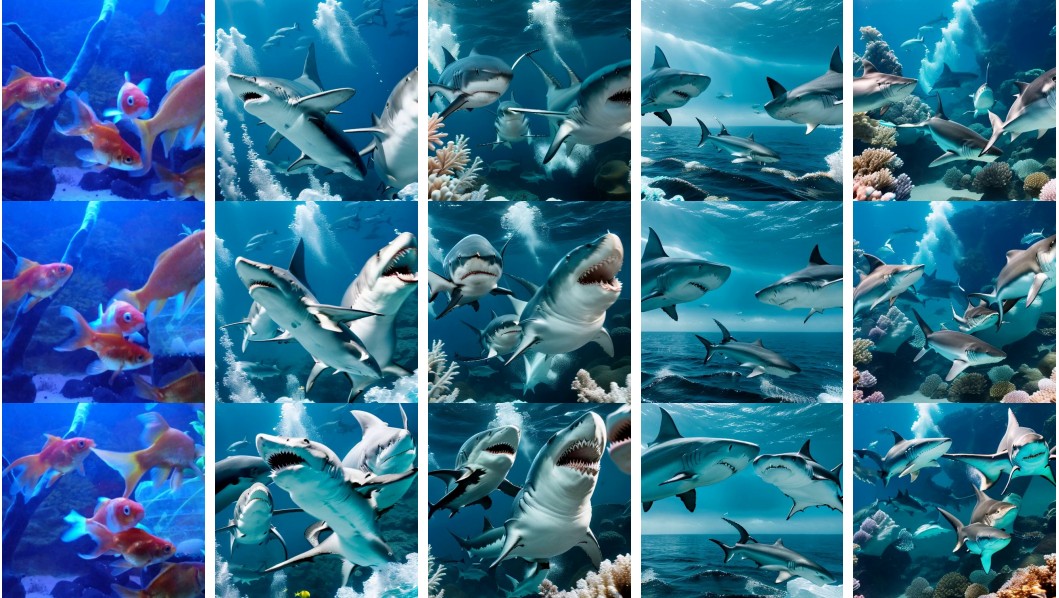

*A group of sharks, swimming, in the ocean, ...*

Figure 11: **Ablation study on applying motion guidance in different "down_block"**.

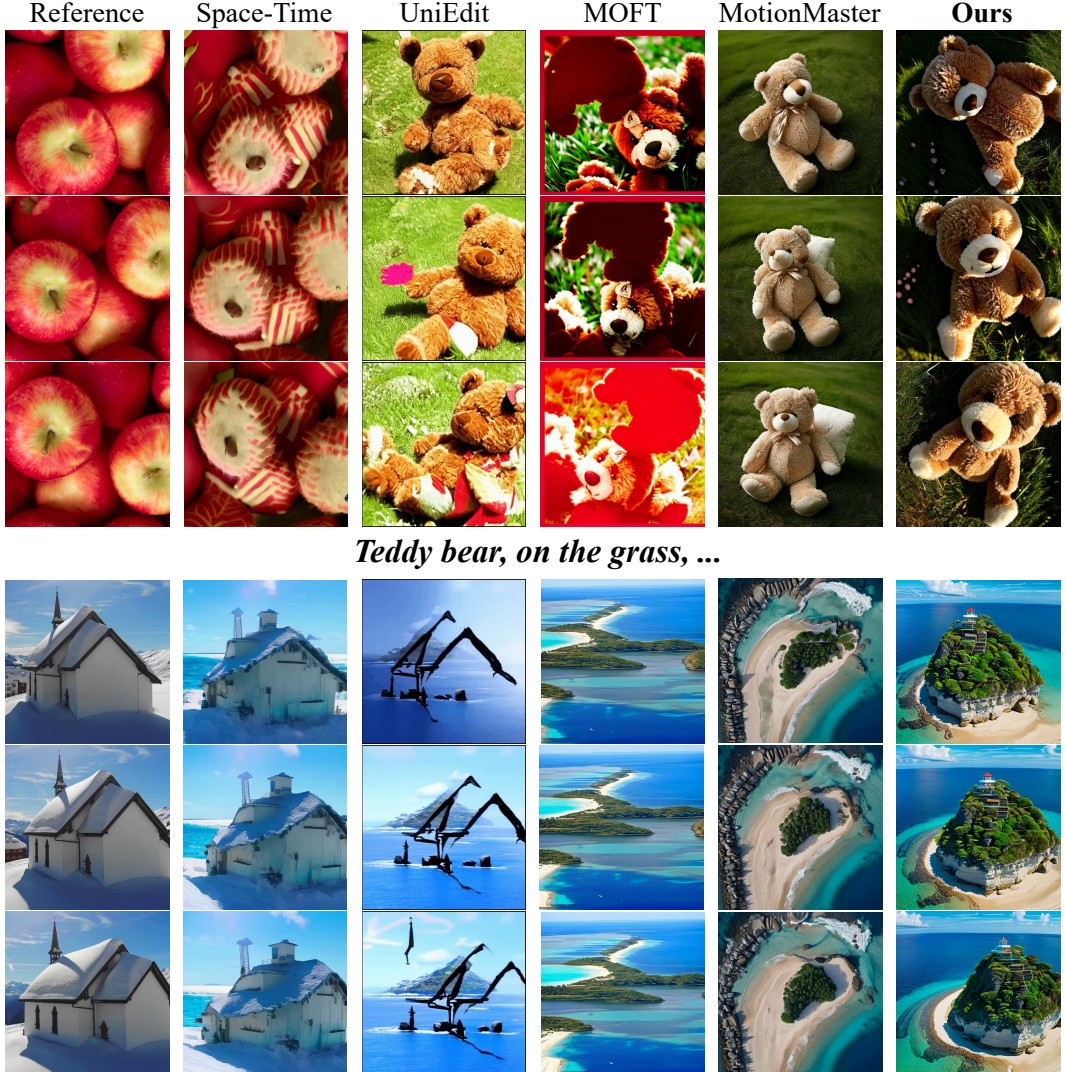

Figure 12: **Comparison with more methods on camera motion cloning**.

| Reference | Space-Time | UniEdit | MOFT | **Ours** |
|-----------|-----------|---------|------|----------|

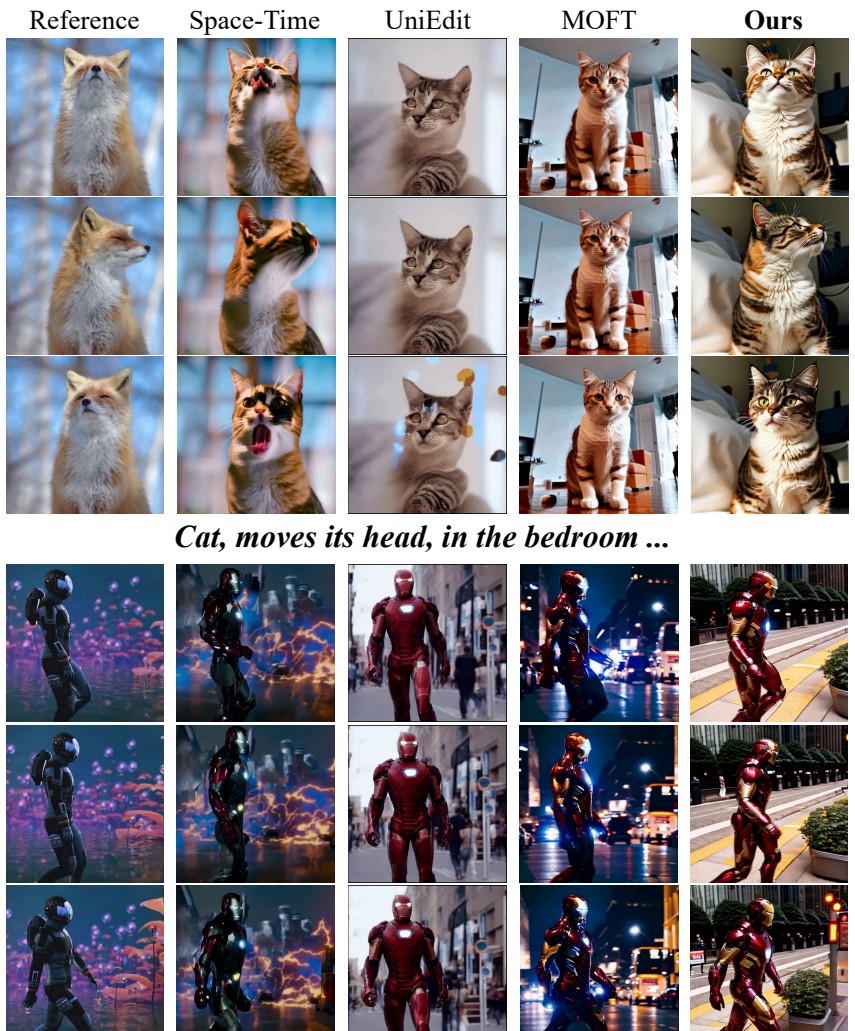

*Cat, moves its head, in the bedroom ...*

*Iron man, walks, in the New York City, ...*

Figure 13: **Comparison with more methods on object motion cloning**.

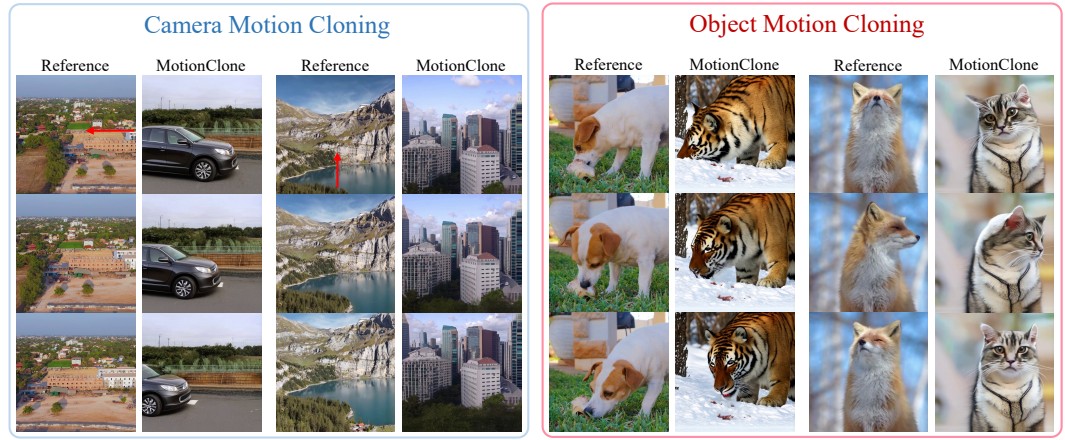

Figure 14: **Results of MotionClone on Diffusion DiT architecture (CogVideoX)**.

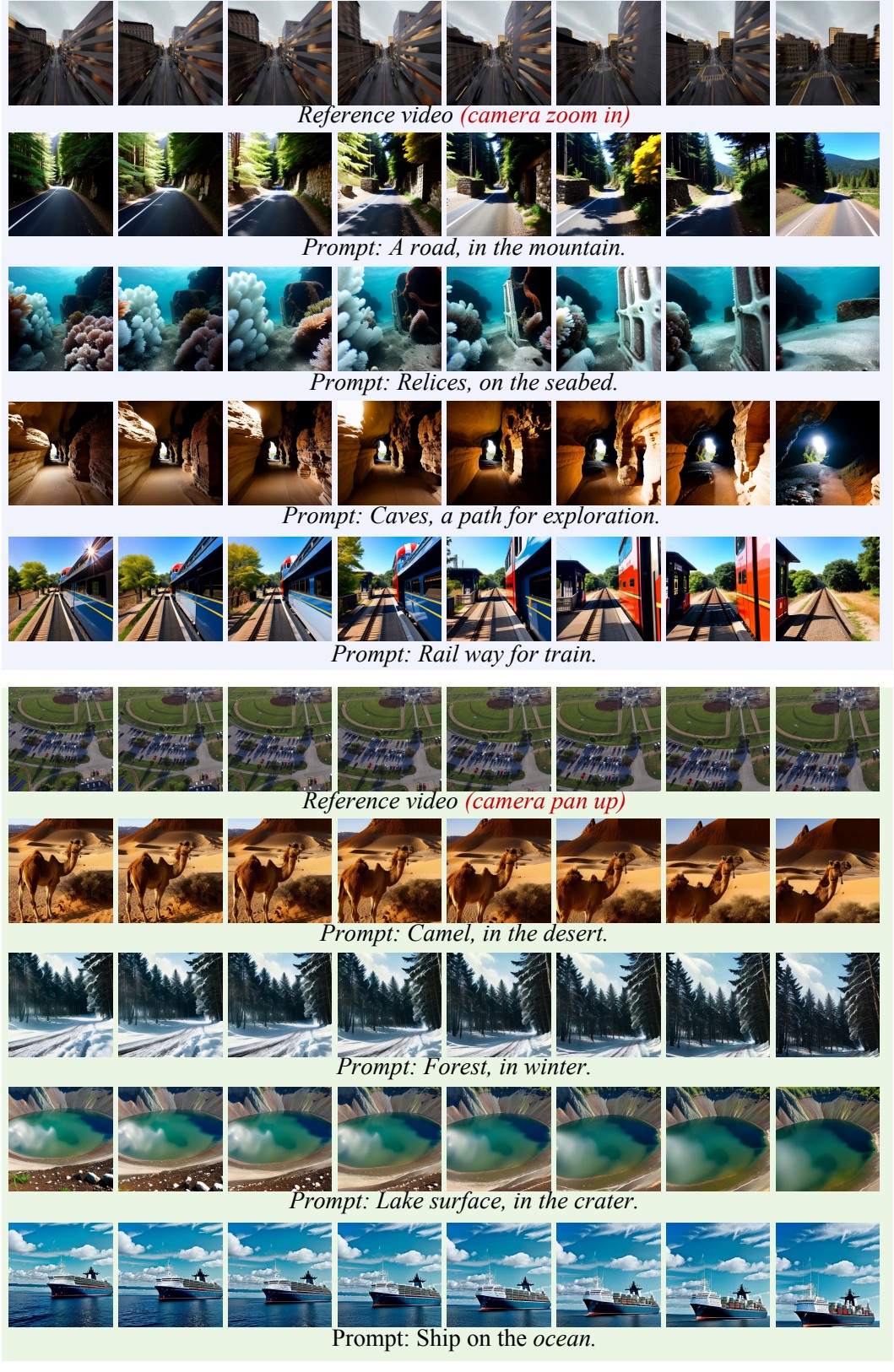

Figure 15: **More results of MotionClone in camera motion cloning**.

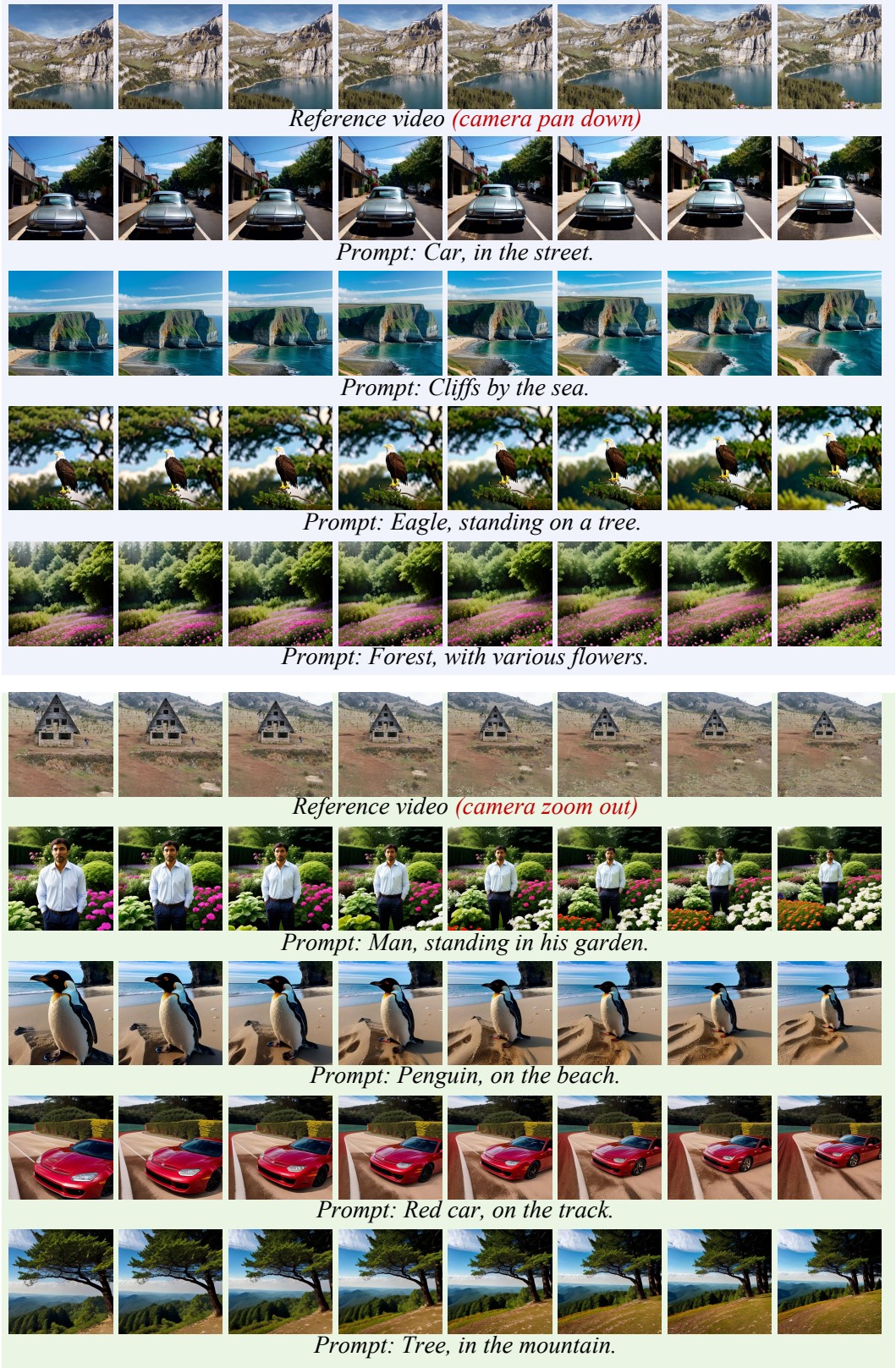

*Reference video (camera pan down)*

*Prompt: Car, in the street.*

*Prompt: Cliffs by the sea.*

*Prompt: Eagle, standing on a tree.*

*Prompt: Forest, with various flowers.*

*Reference video (camera zoom out)*

*Prompt: Man, standing in his garden.*

*Prompt: Penguin, on the beach.*

*Prompt: Red car, on the track.*

*Prompt: Tree, in the mountain.*

Figure 16: **More results of MotionClone in camera motion cloning**.

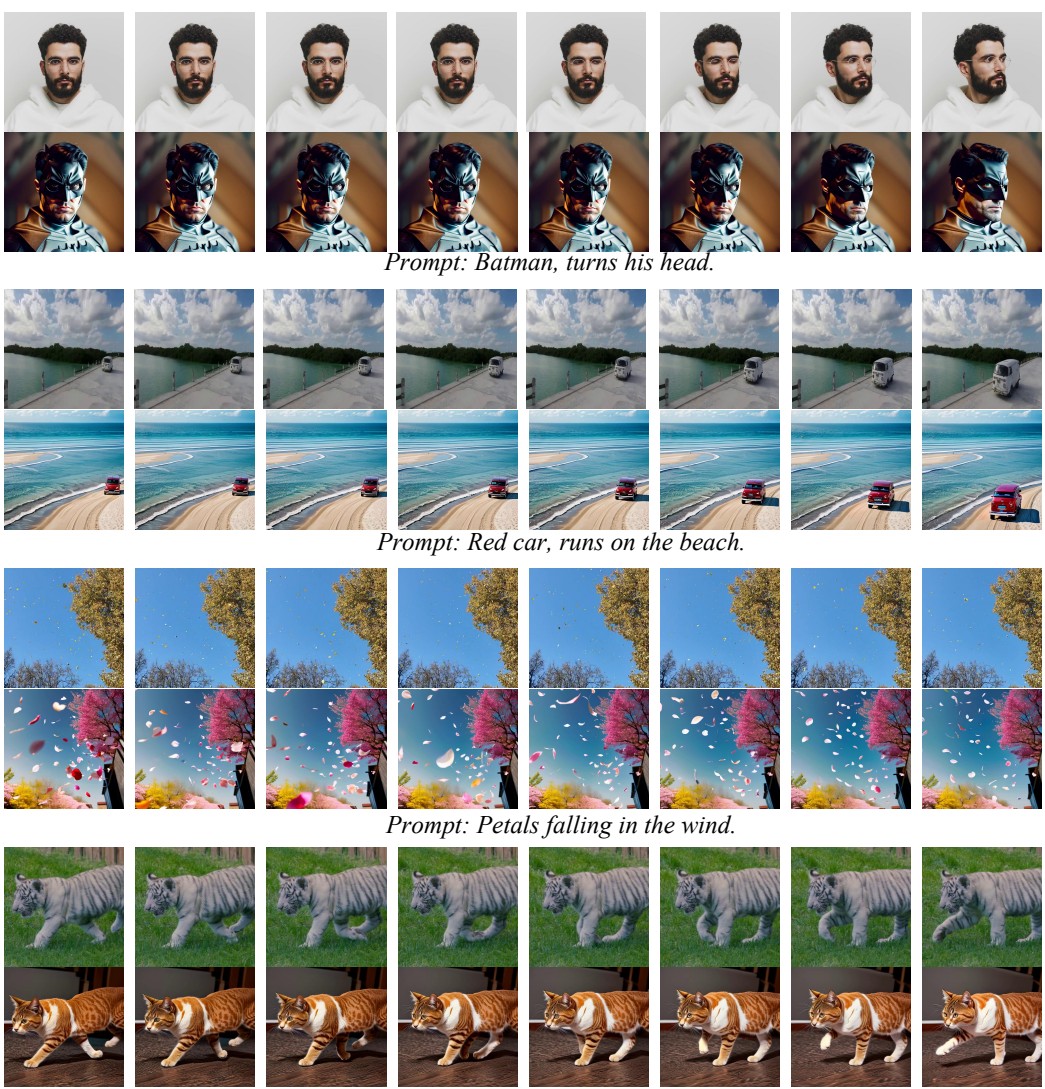

*Prompt: Batman, turns his head.*

*Prompt: Red car, runs on the beach.*

*Prompt: Petals falling in the wind.*

*Prompt: Cat, runs in house.*

Figure 17: **More results of MotionClone in object motion cloning**.

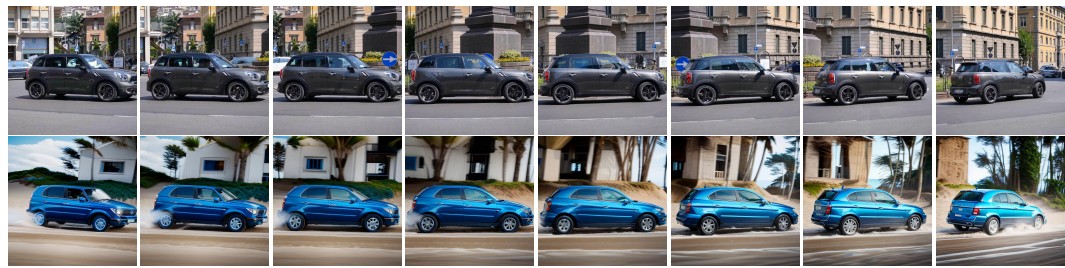

*Prompt: Blue car, runs on the beach.*

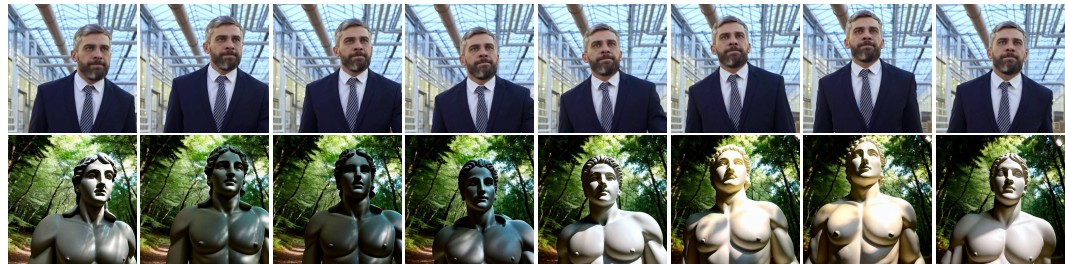

*Prompt: Greek sculpture, walks in the forest.*

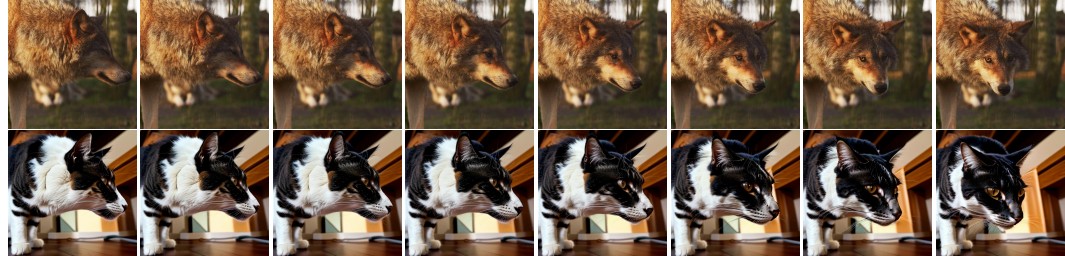

*Prompt: Cat, turns its head in house.*

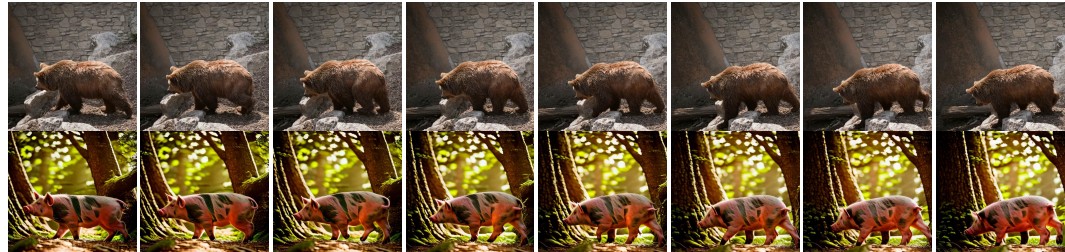

*Prompt: Pig, walks in the forest.*

Figure 18: **More results of MotionClone in object motion cloning**.

