# OpenReview forum: "MotionClone: Training-Free Motion Cloning for Controllable Video Generation"
_ICLR.cc/2025/Conference — ICLR 2025 Poster_

### Official Review · Reviewer_WTAP · 2024-10-31

**Soundness:** 3
**Presentation:** 3
**Contribution:** 2
**Rating:** 6
**Confidence:** 5

**Summary:**

The paper introduces MotionClone, a training-free framework designed for motion cloning in controllable video generation. Unlike traditional methods that require tailored training or fine-tuning, MotionClone leverages the temporal attention mechanism within video generation models to capture and clone motion from reference videos. This approach allows for detailed motion synthesis while minimizing dependencies on the structural components of the reference video, enabling flexible motion cloning across various scenarios. The framework is versatile, supporting tasks like text-to-video (T2V) and image-to-video (I2V), and it has been shown to outperform existing methods in terms of motion accuracy and text alignment.

**Strengths:**

1. MotionClone eliminates the need for complex training processes, making it more accessible and efficient.
2. The framework supports multiple video generation tasks, including T2V and I2V, demonstrating its broad applicability.
3. Enhanced Motion Accuracy: MotionClone outperforms existing methods in capturing and cloning localized object motions with higher fidelity.

**Weaknesses:**

The core idea of this paper, i.e., achieving motion cloning by constraining the temporal attention map, has already appeared in previous work, specifically in UniEdit [1] and MotionMaster [2]. The authors should acknowledge this prior work as the source rather than presenting it as an original contribution.

The primary technical distinction from [1][2] is the proposal to use a sparse temporal mask
M rather than constraining all temporal attention maps. However, this choice lacks motivation or insight, which makes its effectiveness seem questionable.

Although the authors call this approach “motion cloning,” it is functionally similar to previous video editing methods [1][3], which also replace appearance while preserving motion. The authors should compare their method with this class of approaches. It is noteworthy that UniEdit [1] and MotionMaster [2] appear capable of achieving the same objectives as this work, namely cloning object and camera motion.

As mentioned in (1), the proposed method closely resembles [1][2]. Therefore, these methods should also be included in the comparisons. Moreover, It is recommended that the authors compare FID scores of generated videos to evaluate video quality.

[1] Bai J, He T, Wang Y, et al. Uniedit: A unified tuning-free framework for video motion and appearance editing[J]. arXiv preprint arXiv:2402.13185, 2024.
[2] Hu T, Zhang J, Yi R, et al. MotionMaster: Training-free Camera Motion Transfer For Video Generation[J]. arXiv preprint arXiv:2404.15789, 2024.
[3] Qi C, Cun X, Zhang Y, et al. Fatezero: Fusing attentions for zero-shot text-based video editing[C]//Proceedings of the IEEE/CVF International Conference on Computer Vision. 2023: 15932-15942.

**Questions:**

Please refer to the Weaknesses section.

---

> ### Author Response · Authors · 2024-11-21
> **Response to Reviewer WTAP (1/2)**
>
> We are very encouraged to see that you found our work " more accessible and efficient". We sincerely thank you for your valuable suggestions, which certainly help improve our work. We have accordingly refined our paper as follows:
>
> **Q1: Achieving motion cloning by constraining the temporal attention map has been explored in UniEdit and MotionMaster.**
>
> **A1:** Thanks for your constructive comment. Both UniEdit[1] and MotionMaster[2] are indeed pioneers in employing temporal attention maps for motion control. However, the main contribution of the proposed MotionClone is that it reveals the dominant components in temporal attention maps drive motion synthesis, while the rest mainly capture  scene-specific noise or very subtle motions. Based on this observation, a primary motion control strategy is designed to perform substantial motion guidance over sparse temporal attention maps, allowing for efficient motion transfer across scenarios. Furthermore, MotionClone demonstrates the effectiveness of employing motion representation extracted from a certain time step, which only performs a single denoising step without time-costing DDIM inversion.  Meanwhile, MotionClone demonstrates its capability to execute motion transfer across scenarios with substantial structure differences, which is particularly crucial for camera motion transfer, as illustrated in Fig. 20 of the updated manuscript.
> In comparison, both UniEdit and MotionMaster directly copy vanilla temporal attention maps for motion injection and pose significant reliance on DDIM inversion, which produces limited motion transfer and demands higher computational consumption in practical deployment. We will include these two methods in the introduction and make a detailed description of the difference between them.
>
> Moreover, as suggested by Reviewer 3PSB, experiments have demonstrated the effectiveness of MotionClone in Diffusion DiT by using CogVideoX[3] (results are presented in Fig. 15 of the updated manuscript), which further validates its potential and generalizability.
>
>
> **Q2: The motivation or insight of the proposed MotionClone.**
>
> **A2:** Thanks for your valuable comment. As shown in Fig. 2 in the manuscript, it is observed that primary control over sparse temporal attention maps produces enhanced motion alignment than plain control over complete temporal attention maps. Motivated by this observation, we postulate that this is because not all temporal attention weights are essential for motion synthesis, with some reflecting scene-specific noise or extremely small motions. Indiscriminate alignment with the entire temporal attention maps dilutes critical motion guidance, resulting in suboptimal motion cloning in novel scenarios. To validate our assumption, we have visualized the mean intensity of the sparse attention map in Fig. 3 in the manuscript, in which the sparse attention map effectively indicates the area and magnitude of motion and suppresses the interference from motion-independent structures, demonstrating its potential for motion guidance. Therefore, we propose MotionClone, a training-free motion cloning solution for controllable video generation, which demonstrates efficiency by avoiding cumbersome inversion processes and offers versatility across various video generation tasks, establishing itself as a highly adaptable and efficient tool for motion customization.

---

> > ### Author Response · Authors · 2024-11-21
> > **Response to Reviewer WTAP (2/2)**
> >
> > **Q3: Adding comparison with video editing methods, such as UniEdit and MotionMaster.**
> >
> > **A3:** Thanks for your valuable suggestion. It is essential to clarify that MotionClone is not a video editing work, as it is capable of handling motion transfer for objects with significant geometric differences (as shown in Fig. 20(a) of the updated manuscript) and structure-independent camera motion transfer (as shown in Fig. 20(b) of the updated manuscript), which are challenging for video editing methods like FateZero[4] and UniEdit[1]. We have supplemented the comparison with more methods, including the suggested UniEdit[1], and MotionMaster[2], and two extra methods recommended by other reviewers, i.e., MOFT[5] and Space-Time Diffusion Features[6]. The qualitative results are presented in Figs. 16-17 in the updated manuscript and the quantitative results are listed below. Since MotionMaster can only handle camera motion, it is not included in the quantitative comparison with both camera and object motion.
> >
> > |Method               | MOFT   | Space-Time | UniEdit | MotionClone     |
> > | :-----|:----: |:----: |:----: |:----: |
> > |Textual Alignment    | 0.2982 | 0.2452     | 0.3083  | **0.3187** |
> > |Temporal Consistency | 0.9587 | 0.9531     | 0.9508  |  **0.9621** |
> >
> > As presented in Figs. 16-17 of the updated manuscript,  the proposed MotionClone achieves better motion alignment while performing effective appearance preservation, thus offering superior motion customization. Such advantages are also validated in the above tab, in which MotionClone gains the first score in both textual alignment and temporal consistency.
> >
> > Moreover, it is worth mentioning that all of the compared methods require time-consuming DDIM inversion, which implies high computation costs in practical application. In comparison, the proposed MotionClone can extract motion representation from a single denoising step with a unified null-text prompt, offering both efficiency and flexibility.
> >
> > **Q4: It is recommended that the authors compare FID scores of generated videos to evaluate video quality.**
> >
> > **A4:** Thanks for your constructive suggestion. The FID scores of each method are listed below (except MotionMaster which only handles camera motion).
> >
> > |Method | VMC  |  VideoComposer  |  Gen-1   |  Tune-A-Vidoe  |  Control-A-Video  |  MOFT    |  Space-Time  |  UniEdit   |  MotionClone  |
> > | :-----|:----: |:----: |:----: |:----: |:----: |:----: |:----: |:----: |:----: |
> >  | FID     |  193  |  203.06         |  190.13  |  188.99        |  189.06           |  185.43  |  162.15      |  200.35       |    **159.90**         |
> >
> >
> > **Reference**
> >
> > [1] Bai, Jianhong, et al. "Uniedit: A unified tuning-free framework for video motion and appearance editing." arXiv preprint arXiv:2402.13185 (2024).
> >
> > [2] Hu, Teng, et al. "MotionMaster: Training-free Camera Motion Transfer For Video Generation." arXiv preprint arXiv:2404.15789 (2024).
> >
> > [3] Yang, Zhuoyi, et al. "Cogvideox: Text-to-video diffusion models with an expert transformer." arXiv preprint arXiv:2408.06072 (2024).
> >
> > [4] Qi, Chenyang, et al. "Fatezero: Fusing attentions for zero-shot text-based video editing." Proceedings of the IEEE/CVF International Conference on Computer Vision. 2023.
> >
> > [5] Xiao, Zeqi, et al. "Video Diffusion Models are Training-free Motion Interpreter and Controller." arXiv preprint arXiv:2405.14864 (2024).
> >
> > [6] Yatim, Danah, et al. "Space-time diffusion features for zero-shot text-driven motion transfer." Proceedings of the IEEE/CVF Conference on Computer Vision and Pattern Recognition. 2024.

---

> > > ### Comment · Reviewer_WTAP · 2024-11-22
> > > **For Authors**
> > >
> > > My concerns have been largely addressed, thus I am increasing my score from 5 to 6.

---

> > > > ### Author Response · Authors · 2024-11-22
> > > >
> > > > Dear Reviewer WTAP,
> > > >
> > > > Thanks for your positive feedback on our work!
> > > >
> > > > I'm glad to hear that our rebuttal has addressed your concerns. We sincerely appreciate your time and effort in providing such meticulous reviews and constructive suggestions.
> > > >
> > > > Best regards,
> > > >
> > > > Authors

---

### Official Review · Reviewer_eHkX · 2024-11-01

**Soundness:** 3
**Presentation:** 3
**Contribution:** 3
**Rating:** 6
**Confidence:** 4

**Summary:**

The authors propose a training-free framework that transfers motion from reference videos to video generation. They find temporal-attention maps drives motion synthesis. Based on this observation, they utilize this temporal-attention maps as motion guidance to control the motion of generated videos. In addition, they conduct extensive experiments to validate the effectiveness of the proposed method.

**Strengths:**

+ This paper find temporal-attention maps contain important motion information. The temporal-attention map alignment between reference videos and generated videos can transfer motion from reference videos to video generation. This observation is interesting.
+ Based on this observation, the authors propose a simple but effective method, which can transfer global motion into video generation. For efficiency, they also introduce fixed-time motion representation as the motion guidance. It is a practical solution for deployment.
+ The authors conduct extensive experiments to validate the proposed method. Compared with existing methods, the proposed method achieves superior performance on motion-based controllable video generation.

**Weaknesses:**

- Ablation results. The experiments are exhaustive, but they are only the qualitative results. It is necessary to show the quantitative results for each component. With quantitative results, the proposed method is more convinced.
- Temporal attention block. To my knowledge, down blocks also have temporal attention blocks. Thus, it is an option to apply the proposed motion guidance on down blocks. However, we only show the results with motion guidance applied in up blocks. It is necessary to provide the ablation results of temporal attention in down blocks.
- Guidance weight $\lambda$. The value of $\lambda$ is 2000. To my knowledge, it is too large. What the explanation of this high value is?
- The illustration of Figure 4. The Figure 4 needs to be improved. It is difficult for readers to learn the proposed method clearly only with the contents of Figure 4. Besides, there are some errors in Figure 4. For example, motion representation is not $t_{\alpha}$. According to main text, $t_{\alpha}$ is a time step.

**Questions:**

The authors need to conduct more experiments and improve Figure 4, see weaknesses.

---

> ### Author Response · Authors · 2024-11-21
> **Response to Reviewer eHkX (1/2)**
>
> We are very encouraged to see that you found our work simple but effective and identified it as a practical solution for deployment. We sincerely thank you for your valuable suggestions, which certainly help improve our work. We have accordingly refined our paper as follows:
>
> **Q1: Ablation results. The experiments are exhaustive, but they are only the qualitative results. It is necessary to show the quantitative results for each component.**
>
> **A1:** Thanks for your valuable suggestion. We have supplemented the quantitative results of the ablation study, i.e., the CLIP similarity with textual embedding for prompt-following evaluation and the optical-flow based consistency between the generated videos and the reference videos for motion alignment assessment (based on the suggestion of Reviewer gxZm). The detailed results are listed below.
>
> | Setting           | Plain control | $t_\alpha = 800$ | $t_\alpha = 600$ | $t_\alpha = 200$  | Inversion.1 | Inversion.2 | Precise prompt | MotionClone |
> |:-----|:----: |:----: |:----: |:----: |:----: |:----: |:----: |:----: |
> |Textual Alignment ($\uparrow$) | 0.3225                 | 0.2853     | 0.3048              | 0.3085          |  0.2994                | 0.3206                |  0.3199             | 0.3187            |
> | Motion Alignment ($\downarrow$)  | 6.5689                 | 4.7606        | 2.9917          | 3.5687           |  3.1490                |  2.7949                |  2.8123              |  2.8012           |
>
>
> It is observed that plain control over the full temporal attention map generates poor motion alignment. Meanwhile, MotionClone ($t_\alpha = 400$)  is able to produce better motion customization compared to other time steps. Inversion.1 (time-dependence representation from DDIM inversion) fails behind Inversion.2 (representation in $t_\alpha = 400$ from DDIM inversion), which validates that the latter can provide more substantial and consistent motion guidance. Further, it is observed that precise prompt has little effect on video quality.
>
> **Q2: It is necessary to provide the ablation results of temporal attention in down blocks.**
>
> **A2:** Thanks for your constructive suggestion. We have evaluated the performance of applying motion guidance in down blocks, the qualitative results are represented in Fig. 19 of the updated manuscript, and the quantitative results are listed below.
>
> |Setting           | Down\_block.1 | Down\_block.2 | Down\_block.3 | MotionClone (Up\_block.1)  | Up\_block.2 | Up\_block.3 |
> |:-----|:----: |:----: |:----: |:----: |:----: |:----: |
> | Textual Alignment ($\uparrow$) |  0.2993         |  0.3014         |   0.3077        | 0.3187           |   0.3112        | 0.3072 |
> | Motion Alignment ($\downarrow$)  | 8.9487         |  8.9603         | 9.4297         | 2.8012            |  2.8832         | 3.8379 |
>
>
> It can be observed that "up\_ block.1" showcases superiority in motion customization, which performs better motion alignment while enabling effective semantic preservation. We conjecture this is because the video motion is determined by the structure of each frame, which is mainly modeled in the "up\_ block.1", as validated by previous image structure customization work (e.g., FreeControl [1]).
>
> However, it is worth noting that "up\_ block.1" is not necessary for MotionClone to apply motion guidance. As suggested by Reviewer 3PSB, we have validated the effectiveness of MotionClone in Diffusion DiT architecture like CogVideoX[2] (results are presented in Fig. 15 of the updated manuscript), in which we employ an auxiliary attention map in the self-attention module for motion representation. Therefore, MotionClone is expected to work in modules containing rich motion features.

---

> > ### Author Response · Authors · 2024-11-21
> > **Response to Reviewer eHkX (2/2)**
> >
> > **Q3: Why the guidance weight $\lambda$ use high value?**
> >
> > **A3:** Thanks for your valuable comment. The guidance weight $\lambda$ adopts a high value because motion guidance is applied in the temporal attention map, which generates small gradient values in noisy latent input. The detailed proof is provided below.
> >
> > The motion guidance on the sparse temporal attention map can be expressed as:
> >
> > $$
> > g = \left \| \mathcal{M}^{t_\alpha}\cdot(\mathcal{A} _ {ref}^{t_\alpha} -\mathcal{A} _ {gen}^{t}) \right \| _2^2.
> > $$
> >
> > Without loss of generality, considering the motion guidance in a position $(p,i,j)$ indicated by mask $\mathcal{M}^{t_\alpha}$, we have
> >
> > $$
> > g_{p,i,j} = \left \| [\mathcal{A} _ {ref}^{t_\alpha}] _ {p,i,j} -[\mathcal{A} _ {gen}^{t}] _ {p,i,j} \right \| _2^2.
> > $$
> >
> > The $[\mathcal{A} _ {gen}^{t}] _ {p,i,j}$ is calculated by softmax operation in the last frame axis, i.e.,
> >
> > $$
> > [\mathcal{A} _ {gen}^{t}] _ {p,i,j} = \frac{e^{z _ {p,i,j}}}{\sum_{k=1}^{f}e^{z _ {p,i,k}}},
> > $$
> >
> > where $f$ is the frame number. Mathematical, the gradient of $g _ {p,i,j}$ over $z _ {p,i,k}$ can be obtained as:
> >
> > $$
> > \frac{\partial g _ {p,i,j}}{\partial z _ {p,i,k}} = -2\left \| [\mathcal{A} _ {ref}^{t _ \alpha}] _ {p,i,j} -[\mathcal{A} _ {gen}^{t}] _ {p,i,j} \right \|\frac{\partial[\mathcal{A} _ {gen}^{t}] _ {p,i,j}}{\partial z _ {p,i,k}}.
> > $$
> >
> > Furthermore, we have
> >
> > $$
> > \frac{\partial[\mathcal{A} _ {gen}^{t}] _ {p,i,j}}{\partial z _ {p,i,k}} = \frac{(e^{z _ {p,i,j}}\cdot \sum_{k=1}^{f}e^{z _ {p,i,k}})- e^{z _ {p,i,j}}\cdot e^{z _ {p,i,k}}}{(\sum _ {k=1}^{f}e^{z _ {p,i,k}})^2} = [\mathcal{A} _ {gen}^{t}]_{p,i,j}\cdot (1- [\mathcal{A} _ {gen}^{t}] _ {p,i,k}) \ \ if \ \ j =k,
> > $$
> >
> > $$
> > \frac{\partial[\mathcal{A} _ {gen}^{t}] _ {p,i,j}}{\partial z _ {p,i,k}} = \frac{-e^{z _ {p,i,j}}\cdot e^{z _ {p,i,k}}}{(\sum _ {k=1}^{f}e^{z_{p,i,k}})^2} = -[\mathcal{A} _ {gen}^{t}] _ {p,i,j} \cdot [\mathcal{A} _ {gen}^{t}] _ {p,i,k} \ \ if \ \ j \ne k,
> > $$
> >
> > Finally, we can obtain
> > $$
> > \frac{\partial g _ {p,i,j}}{\partial z _ {p,i,k}} = -2\left \| [\mathcal{A} _ {ref}^{t _ \alpha}] _ {p,i,j} -[\mathcal{A} _ {gen}^{t}] _ {p,i,j} \right \|[\mathcal{A} _ {gen}^{t}] _ {p,i,j}\cdot (1- [\mathcal{A} _ {gen}^{t}] _ {p,i,k}) \ \ if \ \ j =k
> > $$
> > $$
> > \frac{\partial g _ {p,i,j}}{\partial z _ {p,i,k}} = 2\left \| [\mathcal{A} _ {ref}^{t_\alpha}] _ {p,i,j} -[\mathcal{A} _ {gen}^{t}] _ {p,i,j} \right \|[\mathcal{A} _ {gen}^{t}] _ {p,i,j} \cdot [\mathcal{A} _ {gen}^{t}] _ {p,i,k} \ \ if \ \ j \ne k,
> > $$
> > Note that the range of attention map values $\mathcal{A}$ is $(0,1)$, thus the value of $\frac{\partial g _ {p,i,j}}{\partial z _ {p,i,k}}$ could be very small, which requires a high weight $\lambda$ to provide sufficient gradient for motion guidance.
> >
> > **Q4: The illustration of Figure 4 needs to be improved.**
> >
> > **A4:** Thanks for your kind reminder. The symbol typo in Fig. 4 on the original version was caused by some inexplicable image compilation problems.  We have corrected them and accordingly updated the figure for better demonstration. The updated figure is represented in the Fig. 4 of the updated manuscript.
> >
> > **Reference**
> >
> > [1] Mo, Sicheng, et al. "Freecontrol: Training-free spatial control of any text-to-image diffusion model with any condition." Proceedings of the IEEE/CVF Conference on Computer Vision and Pattern Recognition. 2024.
> >
> > [2] Yang, Zhuoyi, et al. "Cogvideox: Text-to-video diffusion models with an expert transformer." arXiv preprint arXiv:2408.06072 (2024).

---

> > > ### Comment · Reviewer_eHkX · 2024-11-22
> > >
> > > Thanks authors for the responses. My concerns have been solved. I prefer to keep my score as 6.

---

> > > > ### Author Response · Authors · 2024-11-22
> > > >
> > > > Dear Reviewer eHkX,
> > > >
> > > > Thanks for your positive feedback on our work!
> > > >
> > > > I'm glad to hear that our rebuttal has addressed your concerns. We sincerely appreciate your time and effort in providing such meticulous reviews and insightful comments.
> > > >
> > > > Best regards,
> > > >
> > > > Authors

---

### Official Review · Reviewer_gxZm · 2024-11-02

**Soundness:** 3
**Presentation:** 3
**Contribution:** 2
**Rating:** 6
**Confidence:** 4

**Summary:**

This paper introduces MotionClone, a training-free framework for motion cloning from reference videos, facilitating motion-controlled video generation for tasks like text-to-video and image-to-video. By utilizing sparse temporal attention weights as motion representations, MotionClone enhances motion transfer across different scenarios. It simplifies the process by allowing direct extraction of motion representations in a single denoising step, improving efficiency. Experimental results demonstrate that MotionClone excels in global camera motion and local object motion, achieving high motion fidelity, textual alignment, and temporal consistency.

**Strengths:**

1. The paper is well-written and presents a novel method to perform temporally-consistent video editing in a zero-shot manner.
2. The proposed motion control strategy is simple yet effective through motion guidance from sparse temporal attention map.
3. The paper conducts comphresive experiments to verify the superiority in global camera motion and local object action.

**Weaknesses:**

1. This paper only employs CLIP scores to evaluate the temporal consistentcy of generated videos, which are not convincing. Please provide appropriate metrics like warping error to improve the soundness of MotionClone.
2. MOFT [1] and Space-Time Diffusion Features [2] also performs zero-shot video editing using pre-trained video diffusion models. The proposed method shares similar ideas with them, could the authors give elaborate comparisons with them?

[1] Xiao, Zeqi, Yifan Zhou, Shuai Yang, and Xingang Pan. "Video Diffusion Models are Training-free Motion Interpreter and Controller." arXiv preprint arXiv:2405.14864 (2024).
[2] Yatim, Danah, Rafail Fridman, Omer Bar-Tal, Yoni Kasten, and Tali Dekel. "Space-time diffusion features for zero-shot text-driven motion transfer." In CVPR 2024.

**Questions:**

1. The supplementary materials only show the generated videos of the proposed method. Could the authors provide the comparisons with other methods?
2. Both VMC and the proposed method employs pre-trained video diffusion models to perform zero-shot video editing. The authors are encouraged to fair compare them under the same setting.
3. Section 4.7 shows the limitation and failure cases of the proposed method. Could the authors discuss their reasons and potential solutions?

---

> ### Author Response · Authors · 2024-11-21
> **Response to Reviewer gxZm (1/2)**
>
> We are very encouraged to see that you found our motion control strategy simple yet effective. We sincerely thank you for your valuable suggestions, which certainly help improve our work. We have accordingly refined our paper as follows:
>
> **Q1: This paper only employs CLIP scores to evaluate the temporal consistency of generated videos, which are not convincing. Please provide appropriate metrics like warping error to improve the soundness of MotionClone.**
>
> **A1:** Thanks for your valuable comment. Given motion of real videos is both diverse and temporally consistent, we use the optical-flow based consistency between generated videos and real reference videos for evaluating motion alignment. Meanwhile, considering that the generated videos are expected to adjust structure according to textual prompt (e.g., turns a black swan into a duck) and focus on primary motion, we introduce smooth operation on the estimated optical-flow to eliminate structural differences,
> which can be expressed as:
> $$
> \frac{1}{f-1} \sum _ {k=1}^{f-1} \left \| \varphi(\mathcal{L} _ {ref}^{k\rightarrow k+1  })- \varphi(\mathcal{L} _ {gen}^{k\rightarrow k+1}) \right \|_1
> $$
> where $\mathcal{L} _ {ref}^{k\rightarrow k+1}$ is the estimated optical flow from the $k$-th frame to the $(k+1)$-th frame in reference video, and $\mathcal{L} _ {gen}$ refers to the optical flow of generated videos, $f$ is the frame number, and $\varphi(\cdot)$ is the smooth operation to eliminate structural differences (spatial mean filter is adopted in current implementation). We adopt RAFT[1] for optical flow estimation, and the corresponding scores of different methods under different mean filter sizes are listed below.
>
> | Method     |     VMC    | VideoComposer | Gen-1  | Tune-A-Video | Control-A-Video | MOFT   | Space-Time | UniEdit | MotionClone     |
> |:-----|:----: |:----: |:----: |:----: |:----: |:----: |:----: |:----: |:----: |
> | filter size = 5 |  4.9786 |  4.1505        |  4.9010 |  7.3041       |  3.5035          |  3.9819 |  2.8696     |  4.0133  |  **2.8012**|
> | filter size = 7 |  4.9619 |  4.1344        |  4.8824 |  7.2786      |  3.4895          |  3.9668 |  2.8557     |  3.9985  |  **2.7857** |
> | filter size = 9 |  4.9453 |  4.1182        |  4.8639 |  7.2531       |  3.4757          |  3.9518 |  2.8416     |  3.9841  |  **2.7695** |
>
> As presented in the above table, the proposed MotionClone achieves the best score in motion alignment under different mean filter sizes, validating its superiority in achieving effective motion transfer.
>
> **Q2: Add the comparison with MOFT [1] and Space-Time Diffusion Features [2].**
>
> **A2:** We have supplemented the comparison with more methods, including the suggested MOFT[2] and Space-Time Diffusion Features[3], and two extra methods recommended by other reviewers, i.e., UniEdit[4], and MotionMaster[5]. The qualitative results are represented in Figs. 16-17 of the updated manuscript and the quantitative results are listed below. Note that MotionMaster can only handle camera motion, thus it is not included in the quantitative comparison with both camera and object motion.
>
> |Method               | MOFT   | Space-Time | UniEdit | MotionClone |
> | :---        |    :----:   |          :----:   | :----:   | :----:   |
> |Textual Alignment    | 0.2982 | 0.2452     | 0.3083 | **0.3187** |
> |Temporal Consistency | 0.9587 | 0.9531     | 0.9508  | **0.9621**|
>
> As presented in Figs. 16-17 of the updated manuscript,  the proposed MotionClone achieves better motion alignment while performing effective appearance preservation, thus offering superior motion customization. Such advantages are also validated in the above tab, in which MotionClone gains the first score in both textual alignment and temporal consistency.
>
> Moreover, it is worth mentioning that all of the compared methods require time-consuming DDIM inversion, which implies high computation costs in practical application. In comparison, the proposed MotionClone can extract motion representation from a single denoising step with a unified null-text prompt, offering both efficiency and flexibility.
>
>
> **Q3: The supplementary materials only show the generated videos of the proposed method. Could the authors provide the comparisons with other methods ?**
>
> **A3:** Thanks for your constructive suggestion. The videos of the compared methods are provided in the updated supplementary materials.

---

> > ### Author Response · Authors · 2024-11-21
> > **Response to Reviewer gxZm (2/2)**
> >
> > **Q4: Both VMC and the proposed method employ pre-trained video diffusion models to perform zero-shot video editing. The authors are encouraged to fair compare them under the same setting.**
> >
> > **A4:** Thanks for your valuable suggestion. We have transferred VMC from the original Show-1[6] to AnimateDiff[7] by using the provided hyperparameters. As shown in Fig. 18 of the updated manuscript, the motion alignment and prompt-following capabilities of VMC indeed have been improved with the help of a stronger base model but still perform limited motion customization. We conjecture this can be caused by two aspects: i) the original VMC is designed in pixel-space while AnimateDiff works in latent space; and ii) VMC performs fine-tuning the temporal attention layer of video diffusion models, which may cause a performance drop.
> >
> > **Q5: Section 4.7 shows the limitation and failure cases of the proposed method. Could the authors discuss their reasons and potential solutions?**
> >
> > **A5:** Thanks for your valuable comment. For Diffusion U-Net, MotionClone is conducted in "up\_block.1", and the spatial resolution of diffusion features in temporary attention within this decoder block is significantly lower than that of input videos. For example, given a $16\times 512 \times 512$ video input, the spatial resolution of ``up\_block.1" is only $16\times16$, thus MotionClone struggles in handling local subtle motion. This problem can be alleviated by using video inputs with higher resolution, which facilitates increased resolution of motion representation for better local motion rendering.
> > Additionally, when multiple moving objects overlap, MotionClone risks quality dropping, attributing that coupled motion raises the difficulty of motion cloning. This can be relieved by using the initial noise based on reference video, i.e, $z_T^{gen} = \sqrt{\bar{\alpha}_T}z_0^{ref} + \sqrt{1-\bar{\alpha}_T}\epsilon_t$, which helps reduce the initialization gap.
> >
> > Furthermore, following the suggestion of Reviewer 3PSB, MotionClone has demonstrated its effectiveness in Diffusion DiT architecture by using CogVideoX[8] (results are presented in Fig. 15 of the updated manuscript), thus it is expected that better results can be obtained by combining MotionClone with a more advanced T2V base model. For instance, MotionClone can handle up to 49 frames in CogVideoX, unlocking its potential for long video motion customization.
> >
> > **Reference**
> >
> > [1] Teed, Zachary, and Jia Deng. "Raft: Recurrent all-pairs field transforms for optical flow." Computer Vision–ECCV 2020: 16th European Conference, Glasgow, UK, August 23–28, 2020, Proceedings, Part II 16. Springer International Publishing, 2020.
> >
> > [2] Xiao, Zeqi, et al. "Video Diffusion Models are Training-free Motion Interpreter and Controller." arXiv preprint arXiv:2405.14864 (2024).
> >
> > [3] Yatim, Danah, et al. "Space-time diffusion features for zero-shot text-driven motion transfer." Proceedings of the IEEE/CVF Conference on Computer Vision and Pattern Recognition. 2024.
> >
> > [4] Bai, Jianhong, et al. "Uniedit: A unified tuning-free framework for video motion and appearance editing." arXiv preprint arXiv:2402.13185 (2024).
> >
> > [5] Hu, Teng, et al. "MotionMaster: Training-free Camera Motion Transfer For Video Generation." arXiv preprint arXiv:2404.15789 (2024).
> >
> > [6] Zhang, David Junhao, et al. "Show-1: Marrying pixel and latent diffusion models for text-to-video generation." International Journal of Computer Vision (2024): 1-15.
> >
> > [7] Guo, Yuwei, et al. "Animatediff: Animate your personalized text-to-image diffusion models without specific tuning." arXiv preprint arXiv:2307.04725 (2023).
> >
> > [8] Yang, Zhuoyi, et al. "Cogvideox: Text-to-video diffusion models with an expert transformer." arXiv preprint arXiv:2408.06072 (2024).

---

> > > ### Comment · Reviewer_gxZm · 2024-11-22
> > >
> > > Thanks for your elaborate response. My concerns have been well addressed, so I will increase my score to 6.

---

> > > > ### Author Response · Authors · 2024-11-22
> > > >
> > > > Dear Reviewer gxZm,
> > > >
> > > > Thanks for your positive feedback on our work!
> > > >
> > > > I'm glad to hear that our rebuttal has addressed your concerns. We sincerely appreciate your time and effort in providing such meticulous reviews and valuable suggestions.
> > > >
> > > > Best regards,
> > > >
> > > > Authors

---

### Official Review · Reviewer_3PSB · 2024-11-02

**Soundness:** 3
**Presentation:** 3
**Contribution:** 2
**Rating:** 6
**Confidence:** 5

**Summary:**

This paper proposes a training-free method that enables motion cloning from reference videos. It explores the use of sparse temporal attention as guidance for motion cloning, which can be extracted through a single denoising step.  Experiments have been conducted to evaluate the effectiveness both qualitatively and quantitatively.

**Strengths:**

1. The exploration of temporal attention for motion representation is interesting and novel to me.
2. The presented results exhibit good motion cloning.
3. The paper is well-written and easy to follow.

**Weaknesses:**

1. The generalization of the proposed method is a concern. The proposed method heavily relies on temporal attention maps, where spatial and temporal attention are separate. Can the proposed method be applied to the latest video methods, such as CogVideoX, which employ the full attention layers?
2. Some existing methods have explored motion guidance in T2V models, such as [1]. The authors should cite and discuss these works.
3. The generated videos exhibit inconsistent details across different frames, such as the robot's eyes and legs in Fig. 1. A better T2V model might help to resolve these inconsistencies.

[1] Xiao, Zeqi, et al. "Video Diffusion Models are Training-free Motion Interpreter and Controller."

**Questions:**

See weaknesses.

---

> ### Author Response · Authors · 2024-11-21
> **Response to Reviewer 3PSB (1/2)**
>
> We are highly encouraged to see that you found our work interesting and novel. We sincerely thank you for your valuable suggestions, which certainly help improve our work. We have accordingly refined our paper as follows:
>
> **Q1: Can the proposed method be applied to the latest video methods, such as CogVideoX, which employ the full attention layers.**
>
> **A1:** Thanks for your constructive suggestion. We have validated the potential of the proposed MotionClone in the latest DiT-based T2V model, i.e., CogVideoX-2B [1], in which MotionClone demonstrates inspiring effectiveness in training-free motion customization by only using motion representation from a single denoising step, offering both generalizability and flexibility.
> The results are represented in Fig. 15 of the updated manuscript (the corresponding video files are provided in the updated supplementary materials), and the detailed implementations are listed below.
>
> The video synthesis stage in Diffusion DiT involves the collaboration between textual token $s_{text} \in \mathbb{R}^{(1 \times n_1 \times c)}$ and image token $s_{img} \in \mathbb{R}^{(1 \times n_2 \times c)}$, in which $n_1$ and $n_2$ are token number, and $c$ is channel number. During video synthesis, CogVideoX leverages a full attention module for complete relation modeling, in which the resolution of the vanilla attention map thus reaches $1 \times (n_1+n_2) \times (n_1 + n_2)$. Since we focus on the dependence between image tokens for motion transfer across scenarios, we introduce an auxiliary attention map to better describe inter-frame dependencies. Specifically, within the self-attention module, the image token is reshaped into $s_{img} \in \mathbb{R}^{(1 \times n_{h} \times h \times w \times f \times c)}$, in which $n_{h}$ is the head number of multi-head attention, $h$ and $w$ are spatial resolution, and $f$ is frame number, the corresponding attention map  $A_{aux} \in  \mathbb{R}^{1 \times n_{h} \times h \times w \times f \times f}$ can be obtained by using self-attention in the last frame axis. Following the operation in Diffusion U-Net, MotionClone applies sparse constraints $A_{aux}$ according to the rank of their values. For given real reference videos, we use null-text as the default prompt for preparing motion representation to facilitate user-friendly deployment.
>
> **Q2: Some existing methods have explored motion guidance in T2V models, such as [1]. The authors should cite and discuss these works.**
>
> **A2:** Thanks for your valuable suggestion. We have supplemented the comparison with more methods, including the suggested MOFT[2] and three extra methods recommended by other reviewers, i.e., Space-Time Diffusion Features[3], UniEdit[4], and MotionMaster[5]. We will cite all of these compared methods and provide a corresponding introduction to them. The qualitative results are represented in Figs. 16-17 in the updated manuscript and the quantitative results are listed below. Note that MotionMaster can only handle camera motion, thus it is not included in the quantitative comparison with both camera and object motion.
>
> |Method               | MOFT   | Space-Time | UniEdit | MotionClone |
> | :---        |    :----:   |          :----:   | :----:   | :----:   |
> |Textual Alignment    | 0.2982 | 0.2452     | 0.3083 | **0.3187** |
> |Temporal Consistency | 0.9587 | 0.9531     | 0.9508  | **0.9621**|
>
> As presented in Figs. 16-17 of the updated manuscript,  the proposed MotionClone achieves better motion alignment while performing effective appearance preservation, thus offering superior motion customization. Such advantages are also validated in the above tab, in which MotionClone gains the best score in both textual alignment and temporal consistency.
> Moreover, it is worth mentioning that all of the compared methods require time-consuming DDIM inversion, which implies high computation costs in practical application. In comparison, the proposed MotionClone can extract motion representation from a single denoising step with a unified null-text prompt, offering both efficiency and flexibility.
>
> **Q3: The generated videos exhibit inconsistent details across different frames, such as the robot's eyes and legs in Fig. 1. A better T2V model might help to resolve these inconsistencies.**
>
> **A3:** Thanks for your valuable comment. MotionClone aims to provide a training-free motion customization solution for existing video diffusion models, however, its capability is still limited by the adopted base model. Given that MotionClone has demonstrated its effectiveness in Diffusion DiT architecture (i.e., CogVideoX[5]),  it is expected that its performance can be further enhanced with more advanced models. For instance, MotionClone can handle up to 49 frames in CogVideoX, unlocking its potential for long video motion customization.

---

> ### Author Response · Authors · 2024-11-21
> **Response to Reviewer 3PSB (2/2)**
>
> **Reference**
>
> [1] Yang, Zhuoyi, et al. "Cogvideox: Text-to-video diffusion models with an expert transformer." arXiv preprint arXiv:2408.06072 (2024).
>
> [12] Xiao, Zeqi, et al. "Video Diffusion Models are Training-free Motion Interpreter and Controller." arXiv preprint arXiv:2405.14864 (2024).
>
> [3] Yatim, Danah, et al. "Space-time diffusion features for zero-shot text-driven motion transfer." Proceedings of the IEEE/CVF Conference on Computer Vision and Pattern Recognition. 2024.
>
> [4] Bai, Jianhong, et al. "Uniedit: A unified tuning-free framework for video motion and appearance editing." arXiv preprint arXiv:2402.13185 (2024).
>
> [5] Hu, Teng, et al. "MotionMaster: Training-free Camera Motion Transfer For Video Generation." arXiv preprint arXiv:2404.15789 (2024).

---

> > ### Comment · Reviewer_3PSB · 2024-11-22
> > **Official Comment of  Reviewer 3PSB**
> >
> > Thank you to the authors for their efforts. Most of my concerns have been addressed. I will raise my rating to 6.

---

> > > ### Author Response · Authors · 2024-11-22
> > >
> > > Dear Reviewer 3PSB,
> > >
> > > Thanks for your positive feedback on our work!
> > >
> > > I'm glad to hear that our rebuttal has addressed your concerns. We sincerely appreciate your time and effort in providing such meticulous reviews and insightful comments.
> > >
> > > Best regards,
> > >
> > > Authors

---

### Author Response · Authors · 2024-11-21

We sincerely appreciate all reviewers for their time and efforts in the review.
We are highly encouraged that:
Reviewer 3PSB found our work **interesting, novel, and "exhibit good motion cloning"**;
Reviewer gxZm described our work as **novel** and the proposed motion control strategy is **simple yet effective**;
Reviewer eHkX found our observation interesting and the method is **"simple but effective"** and "offers a practical solution for deployment"; Reviewer WTAP described our work **"more accessible and efficient"** and ``demonstrating the broad applicability and enhanced Motion Accuracy".
These positive comments have greatly motivated us, which will certainly guide our future work.

We have carefully revised the manuscript based on the valuable feedback and suggestions. Specifically, we have improved the manuscript from the following aspects:

i) We have validated the potential of the proposed **MotionClone in the latest DiT-based T2V model**, i.e., CogVideoX-2B [1], in which MotionClone demonstrates effectiveness in training-free motion customization by only using motion representation from a single denoising step, offering both generalizability and flexibility.

ii) We have added the **comparison with more methods**, including MOFT [2], Space-Time Diffusion Features[3], UniEdit[4], and MotionMaster[5].

iii) We have introduced **optical-flow based consistency** between generated videos and real reference videos for evaluating motion alignment and added the evaluation regarding the FID metric.

iv) We have further analyzed the **reasons regarding the limitations** and accordingly provided corresponding **potential solutions**.

v) We have supplemented the **quantitative results of the ablation study**.

vi) We have provided the ablation study on **applying motion guidance in different down blocks**.

vii) We have provided **detailed proof about the choice of motion guidance weight $\lambda$**.

viii) We have made a **clearer description of the motivation** and accordingly presented the difference between the proposed method with previous video editing work.

The detailed point-by-point responses are listed in the columns below.

**Reference**

[1] Yang, Zhuoyi, et al. "Cogvideox: Text-to-video diffusion models with an expert transformer." arXiv preprint arXiv:2408.06072 (2024).

[2] Xiao, Zeqi, et al. "Video Diffusion Models are Training-free Motion Interpreter and Controller." arXiv preprint arXiv:2405.14864 (2024).

[3] Yatim, Danah, et al. "Space-time diffusion features for zero-shot text-driven motion transfer." Proceedings of the IEEE/CVF Conference on Computer Vision and Pattern Recognition. 2024.

[4] Bai, Jianhong, et al. "Uniedit: A unified tuning-free framework for video motion and appearance editing." arXiv preprint arXiv:2402.13185 (2024).

[5] Hu, Teng, et al. "MotionMaster: Training-free Camera Motion Transfer For Video Generation." arXiv preprint arXiv:2404.15789 (2024).

---

### Meta-Review · Area_Chair_1qvo · 2024-12-20

**Metareview:**

All reviewers have a positive view on this submission, acknowledging the novelty and interest in exploring temporal attention for motion representation. They find the generated motion cloning results promising and the presentation well-written and clear.
As for the concerns raised by the reviewers, the authors have addressed them during the discussion phase. Considering the novelty, experimental results, and the authors' responsiveness to the reviewers' feedback, the paper is recommended to accepted for publication.

**Additional Comments On Reviewer Discussion:**

Reviewers are actively invovled in the discussion with authors and most concerns of reviewers have been addressed by the authors. Several  reviewers raised the score to be positive.

---

### Decision · Program_Chairs · 2025-01-22

Accept (Poster)